# Nucleocapsid-specific T cell responses associate with control of SARS-CoV-2 in the upper airways before seroconversion

Tabea M. Eser[1,2], Olga Baranov[1,2], Manuel Huth[3,4], Mohammed I. M. Ahmed[1,2], Flora Deák[1,2], Kathrin Held [1,2], Luming Lin[1,2], Kami Pekayvaz [5,6], Alexander Leunig [5,6], Leo Nicolai [5,6], Georgios Pollakis [7], Marcus Buggert [8], David A. Price [9,10], Raquel Rubio-Acero[1], Jakob Reich[1], Philine Falk[1], Alissa Markgraf[1], Kerstin Puchinger[1], Noemi Castelletti[1,2], Laura Olbrich[1,2], Kanika Vanshylla[11], Florian Klein [11,12,13], Andreas Wieser[1,2,14], Jan Hasenauer [3,4,15], Inge Kroidl[1,2], Michael Hoelscher[1,2] & Christof Geldmacher[1,2] ✉

Despite intensive research since the emergence of SARS-CoV-2, it has remained unclear precisely which components of the early immune response protect against the development of severe COVID-19. Here, we perform a comprehensive immunogenetic and virologic analysis of nasopharyngeal and peripheral blood samples obtained during the acute phase of infection with SARS-CoV-2. We find that soluble and transcriptional markers of systemic inflammation peak during the first week after symptom onset and correlate directly with upper airways viral loads (UA-VLs), whereas the contemporaneous frequencies of circulating viral nucleocapsid (NC)-specific CD4+ and CD8+ T cells correlate inversely with various inflammatory markers and UA-VLs. In addition, we show that high frequencies of activated CD4+ and CD8+ T cells are present in acutely infected nasopharyngeal tissue, many of which express genes encoding various effector molecules, such as cytotoxic proteins and IFN-γ. The presence of *IFNG* mRNA-expressing CD4+ and CD8+ T cells in the infected epithelium is further linked with common patterns of gene expression among virus-susceptible target cells and better local control of SARS-CoV-2. Collectively, these results identify an immune correlate of protection against SARS-CoV-2, which could inform the development of more effective vaccines to combat the acute and chronic illnesses attributable to COVID-19.

SARS-CoV-2 has infected more than 600 million people and caused more than 6 million deaths worldwide (https://www.worldometers.info/coronavirus). Vaccines designed primarily to elicit neutralizing antibodies against the spike (S) protein initially attenuated the course of disease and protected against the development of severe COVID-19[1–5]. However, the continual emergence of viral escape variants has undermined this approach, and the ongoing pandemic is now largely driven by strains resistant to vaccine-induced antibody-mediated neutralization[6].

Several reports have indicated a likely role for SARS-CoV-2-specific T cells as key determinants of immune protection against severe COVID-19[7–11]. More directly, antigen-specific memory

CD4[+] T cells in the airways have been shown to protect mice against respiratory coronaviruses after vaccination[12], and depletion studies in rhesus macaques vaccinated with adenovirus-encoded S (Ad26.COV2.S) have implicated CD8[+] T cells as important mediators of viral control after intranasal or intratracheal challenge with SARS-CoV-2[13]. It is also notable that antigen-specific memory CD4[+] T cells in the circulation have been associated with immune protection in humans after influenza virus challenge[14]. In line with these observations, SARS-CoV-2 has been shown to induce tissue-resident memory T cell immunity[15,16], but the precise correlates of early viral control and disease mitigation have nonetheless remained elusive[17].

In this study, we investigated the dynamics of adaptive immune responses in relation to markers of disease severity during acute infection with SARS-CoV-2. Our data provided correlative and mechanistic evidence to indicate that viral nucleocapsid (NC)-specific T cells were the central determinants of immune protection, limiting viral replication in the upper airways and suppressing the attendant inflammatory response. Collectively, these observations revealed a cellular and molecular signature of effective antiviral immunity, with potential implications for the development of next-generation vaccines against COVID-19.

## Results

### Viral loads in the upper airways are highly variable during acute infection with SARS-CoV-2

A total of 37 patients with acute COVID-19 were recruited into this study between May and December 2020 (Fig. 1a). All participants had mild symptoms that did not require hospitalization (Table 1)[18]. Twenty-five of these patients were recruited within the first week of symptom onset (median = 5 days, interquartile range [IQR] = 4–6 days). Upper airways viral loads (UA-VLs) were highly variable during the first week of infection (median = $1.7 \times 10^8$ RNA copies/ml, range = $1.7 \times 10^2$ to $9.8 \times 10^{10}$ RNA copies/ml) (Fig. 1b). IgA and IgG responses against the viral S protein were below the detection threshold in all cases (Supplementary Fig. 1), and only 12% of donors (3/25) had detectable neutralization titers at the time of recruitment (Fig. 1c). In the second week of infection, all patients had lower UA-VLs (median = $2.1 \times 10^3$ RNA copies/ml, range = $4.8 \times 10^0$ to $1.1 \times 10^7$ RNA copies/ml) (Fig. 1b), and SARS-CoV-2 neutralization titers became detectable in 92% of cases (23/25), subsequently peaking during the third week of infection (median $IC_{50} = 165$, IQR = 66–375) (Fig. 1c). Most subjects retained detectable neutralization titers until the last study visit 6 months after symptom onset (Fig. 1c). A similar pattern was observed for antibody responses against the viral NC protein (Supplementary Fig. 1).

Collectively, these data established that UA-VLs peaked during the first week of infection, before the emergence of detectable antibody responses, and varied considerably among individuals with mild COVID-19.

### Nucleocapsid-specific T cell responses correlate inversely with upper airways viral loads during acute infection with SARS-CoV-2

T cell responses against the viral NC and S proteins were measured longitudinally using flow cytometry to detect the intracellular production of IFN-γ. SARS-CoV-2-specific CD4[+] T cells were detected more frequently than SARS-CoV-2-specific CD8[+] T cells (Fig. 2a–e and Supplementary Fig. 2). Area under the curve (AUC) analyses revealed that the overall frequency of SARS-CoV-2-specific CD4[+] T cells was higher than the overall frequency of SARS-CoV-2-specific CD8[+] T cells per day across all time points in the study (P < 0.0001) (Fig. 2f), and in both lineages, the overall frequency of NC-specific T cells was higher than the overall frequency of S-specific T cells per day across all time points in the study (P = 0.0102) (Fig. 2g). Higher frequencies of NC-specific CD4[+] T cells and S-specific CD8[+] T cells were detected in patients versus healthy controls during the first week after symptom onset

(P = 0.0005 for NC, P = 0.0085 for S) (Fig. 2h, i). SARS-CoV-2-specific CD4[+] T cell responses typically peaked during the third week after symptom onset for NC (median = 0.045% of CD4[+] T cells, P = 0.0018) and S (median = 0.023% of CD4[+] T cells, P = 0.0063), whereas SARS-CoV-2-specific CD8[+] T cell responses typically peaked during the fourth week after symptom onset for NC (median = 0.024% of CD8[+] T cells, P = 0.042) and during the third week after symptom onset for S (median = 0.033% of CD8[+] T cells, P = 0.038) (Fig. 2h, i). Of note, 51.1% of patients mounted detectable SARS-CoV-2-specific CD4[+] T cell responses during the first week of infection, and 37.7% of patients mounted detectable SARS-CoV-2-specific CD8[+] T cell responses during the first week of infection (Fig. 2e).

In total, 21% of healthy controls had detectable NC-specific T cell responses, and 52% of healthy controls had detectable S-specific T cell responses (Fig. 2h, i), consistent with previous reports[9,19–21]. To investigate this phenomenon, we measured serological reactivity against the four common cold coronaviruses (CCCVs). Strain-specific antibody responses were detected in most patients for NL63 (80%), OC43 (64%), and HKU1 (68%), whereas only 48% of patients were seropositive for 229E (Supplementary Fig. 3a). Data from healthy controls are shown in Supplementary Fig. 3b. There was no association between the presence of early NC-specific CD4[+] or CD8[+] T cell responses and serological reactivity against CCCVs (Supplementary Fig. 3c).

In further analyses, we found a strong inverse correlation between the overall frequency of circulating NC-specific T cells during the first week after symptom onset and UA-VLs (r = −0.75, P < 0.0001) (Fig. 3a). This association was strongest for NC-specific CD4[+] T cells (r = −0.69, P < 0.0001) but was also significant for NC-specific CD8[+] T cells (r = −0.45, P = 0.02) (Fig. 3a). In contrast, we found no such correlations for S-specific T cells, irrespective of lineage (Fig. 3b). Using a censored linear mixed effects model with random individual effects to control for other potential confounders, we also found that incremental increases in the frequencies of NC-specific but not S-specific CD4[+] and CD8[+] T cells reduced individual UA-VLs (Supplementary Fig. 4). Age and gender did not play a significant role. Importantly, the model also controlled for time after symptom onset in the regression analysis, ensuring the results were independent of any natural decay in the UA-VLs.

Collectively, these findings supported a role for early IFN-γ-expressing NC-specific CD4[+] and CD8[+] T cells as mediators of viral clearance in the upper airways, which could have important implications for the development of more effective vaccines against SARS-CoV-2.

### Nucleocapsid-specific T cell responses correlate inversely with markers of systemic inflammation during acute infection with SARS-CoV-2

Excessive production of various chemokines and cytokines, including CXCL10 and CXCL11, has been linked with the severity of COVID-19[22,23]. Using a 26-plex panel, we found that plasma concentrations of CXCL10 and CXCL11 were significantly elevated during the first week after symptom onset (median = 3922 pg/ml and 97.5 pg/ml, respectively) compared with later time points (P < 0.001 or P < 0.0001) (Fig. 3c and Supplementary Fig. 5). Moreover, plasma concentrations of CXCL10 correlated directly with UA-VLs (r = 0.50, P = 0.01) and inversely with the frequency of circulating NC-specific T cells during the first week after symptom onset (r = −0.43, P = 0.002) (Fig. 3c). Similar correlations were found for CXCL11 (r = 0.65, P = 0.0004 versus UA-VLs; r = −0.43, P = 0.03 versus NC-specific T cells) (Supplementary Fig. 5). Other soluble factors were also upregulated significantly during the first week after symptom onset compared with later time points, including CCL3, CCL19, galectin-9, and MICA (Supplementary Fig. 5). Plasma concentrations of CCL2, CCL19, galectin-9, and MICA correlated directly with UA-VLs during the first week after symptom onset (r > 0.4, P < 0.05), and plasma concentrations of CCL19 and MICA

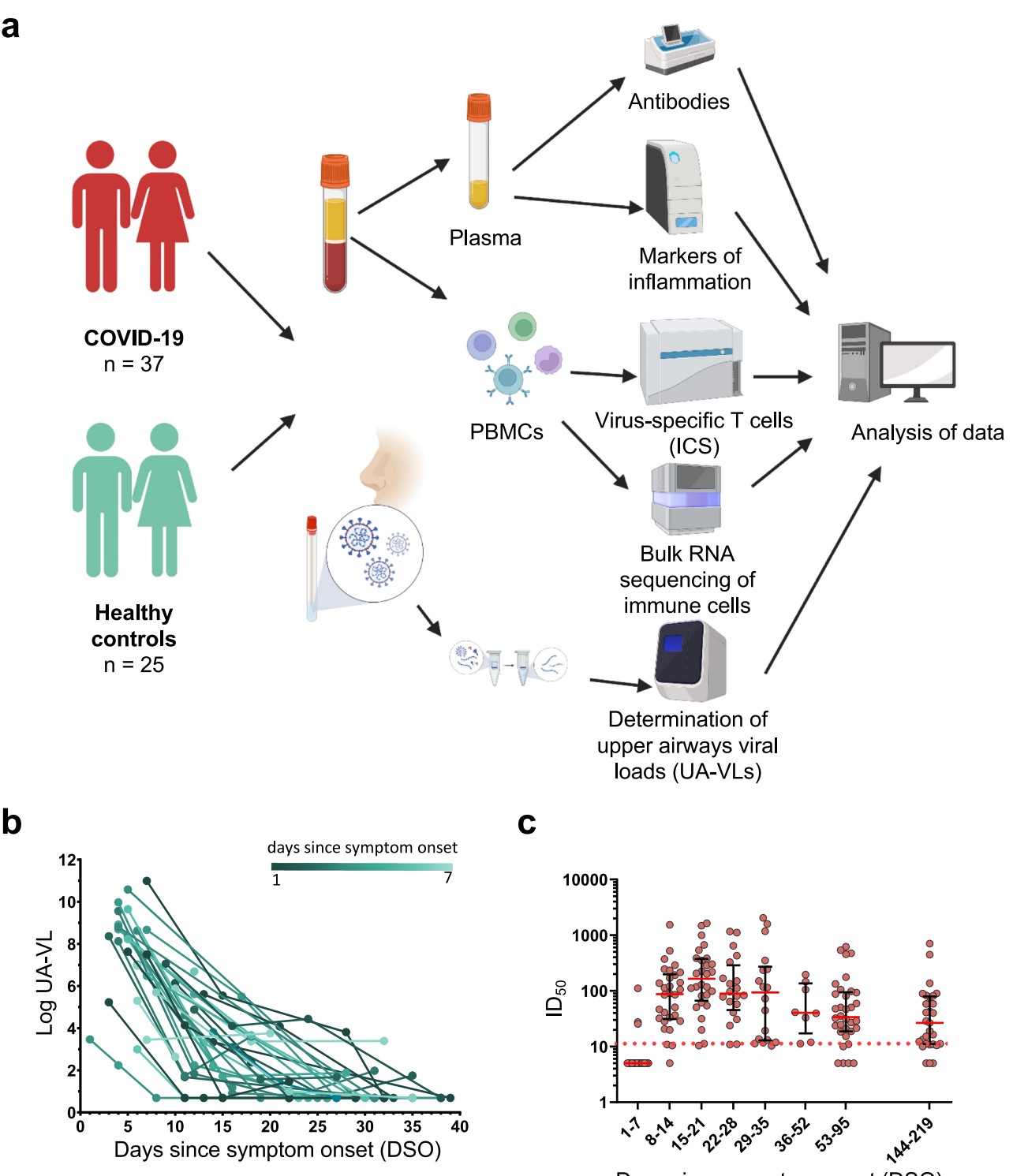

**Fig. 1 | Study overview, upper airways viral loads, and antibody-mediated neutralization of SARS-CoV-2. a** Schematic representation of the study design. Donors were sampled weekly for 1 month and then periodically until 6 months after initial presentation. **b** Longitudinal quantification of upper airways viral loads (UA-VLs) in patients with mild COVID-19 ($n = 25$) recruited during the first week after symptom onset. Each line represents one donor. The green scale stratifies patients according to days since symptom onset at presentation. **c** Pseudovirus neutralization titers ($ID_{50}$) plotted versus days since symptom onset (DSO). Each dot represents one donor at one time point as follows: 0–7 DSO, $n = 25$; 8–14 DSO, $n = 30$; 15–21 DSO, $n = 28$; 22–28 DSO, $n = 20$; 29–35 DSO, $n = 18$; 36–52 DSO, $n = 8$; 53–95 DSO, $n = 34$; 144–219 DSO, $n = 29$. The cutoff is indicated by the dotted red line. Serum samples that did not achieve 50% neutralization ($ID_{50} < 10$) were assigned a value halfway below the lower limit of quantification ($ID_{50} = 5$). Data are shown as median ± IQR. Source data are provided as a source data file. Figure 1a was created with Biorender (publication licence number GL254AMU2N).

## Table 1 | Patient characteristics

| Patients | 37 |
|---|---|
| Gender (female) | 20 [54.5%] |
| Median age (years) [IQR] | 36 [30/49.5] |
| WHO score 1 | 1 [2.7%] |
| WHO score 2 | 14 [37.8%] |
| WHO score 3 | 22 [59.5%] |
| Lung involvement | 21 [56.75%] |
| Recruited within first week after symptom onset | 25 [67.75%] |
| Neutralizing antibodies (1–7 days after symptom onset) | 4 [16%] |
| Anti-Ig nucleocapsid (1–7 days after symptom onset) | 2 [7.6%] |
| Anti-IgA spike (1–7 days after symptom onset) | 0 |
| Anti-IgG spike (1–7 days after symptom onset) | 0 |
| Median log UA-VL (1–7 days after symptom onset) [IQR] | 8.2 [6.9/8.8] |
| Median log UA-VL (8–14 days after symptom onset) [IQR] | 3.3 [1.7/5.03] |

*IQR* interquartile range, *UA-VL* upper airways viral load (RNA copies/ml).

correlated inversely with the frequency of circulating NC-specific T cells during the first week after symptom onset ($r < -0.4$, $P < 0.05$) (Supplementary Fig. 5).

To explore the nature of these associations, we profiled the transcriptomes of circulating immune cell subsets, namely CD4+ T cells, CD8+ T cells, monocytes, and NK cells, isolated during the first week after symptom onset ($n = 14$ patients with mild COVID-19). We initially focused our analysis on previously reported differentially expressed genes (DEGs), notably *STAT1*, *OAS1*, and *EIF2AK2*, which have been implicated in the clearance of SARS-CoV-1 by murine IFN-γ+ NC-specific CD4+ T cells after intranasal vaccination[12]. In our cohort, the frequency of circulating NC-specific CD4+ T cells correlated inversely with gene expression among circulating immune cell subsets for *STAT1* (CD4+ T cells: $r = -0.38$, $P = 0.029$; CD8+ T cells: $r = -0.53$, $P = 0.001$; monocytes: $r = -0.34$, $P = 0.05$; NK cells: $r = -0.39$, $P = 0.023$), *OAS1* (CD4+ T cells: $r = -0.21$, $P = 0.25$; CD8+ T cells: $r = -0.47$, $P = 0.006$; monocytes: $r = -0.60$, $P = 0.0002$; NK cells: $r = -0.5$, $P = 0.003$), and *EIF2AK2* (CD4+ T cells: $r = -0.42$, $P = 0.015$; CD8+ T cells: $r = -0.23$, $P = 0.199$; monocytes: $r = -0.51$, $P = 0.003$; NK cells: $r = -0.43$, $P = 0.012$) (Fig. 4a). Similar correlation trends were observed among the same immune cell subsets for NC-specific CD8+ T cells, and direct correlations were detected for all three markers versus UA-VLs (Fig. 4a).

Next, we conducted mean expression analyses for pathways classified as *Signal Transduction*, *Signaling Molecules and Interaction*, *Immune System*, and *Cell Growth and Death* according to the Kyoto Encyclopedia of Genes and Genomes (KEGG). Correlations were performed against the frequency of circulating NC-specific CD4+ T cells (Fig. 4b), the frequency of circulating NC-specific CD8+ T cells (Fig. 4c), and UA-VLs (Fig. 4d). Signaling pathways involved in the host response and inflammation, including those for NF-κB, RIG-1-like receptors (RLRs), and JAK-STAT, generally correlated inversely with the frequency of NC-specific CD4+ T cells and directly with UA-VLs (Fig. 4b, d). The frequency of circulating NC-specific CD8+ T cells also correlated inversely with the NF-κB pathway but directly with other pathways, including those associated with cytotoxicity (Fig. 4c). The pathway scores were then included in the censored linear mixed effect model for further investigation. These analyses confirmed that the pathway scores for NF-κB and RLR signaling, as well as other pathways, including antigen processing and presentation, for at least one of the immune cell subsets in each case were influenced by UA-VLs (Supplementary Fig. 6).

Unsupervised analyses further revealed three distinct clusters within the overall dataset (Fig. 4e). One group incorporating NC-specific CD4+ T cell responders was characterized predominantly by downregulation of immune system and signaling pathways among

circulating immune cell subsets, whereas another cluster incorporating NC-specific CD4+ T cell non-responders was characterized predominantly by upregulation of immune system and signaling pathways among circulating immune cell subsets (Fig. 4e). The other cluster incorporated a mixed group of NC-specific CD4+ T cell responders and non-responders, in which immune system and signaling pathways among circulating immune cell subsets were either upregulated, predominantly among T cells, or downregulated, predominantly among monocytes and NK cells (Fig. 4e).

Collectively, these data showed that systemic upregulation of inflammatory pathways during early infection was positively associated with high viral burdens in the upper airways and negatively associated with the frequencies of circulating NC-specific CD4+ and CD8+ T cells, which in turn suggested that these immune effectors likely mitigated the inflammatory response via enhanced clearance of SARS-CoV-2.

### T cells in the upper airways express mRNAs encoding IFN-γ and cytotoxic effector molecules during acute infection with SARS-CoV-2

To pursue this line of investigation, which suggested a potential role for tissue-recirculating and/or tissue-resident NC-specific CD4+ and/or CD8+ T cells as mediators of viral control at the site of infection[12], we interrogated two single-cell RNA sequencing datasets available in the public domain. The primary dataset communicated by Ziegler et al. incorporated nasopharyngeal material collected from patients in intensive care with no recent history of COVID-19 ($n = 6$) and patients with mild to severe COVID-19 ($n = 37$)[24]. A total of 32,587 cells were analyzed in the original study and annotated to 32 clusters spanning distinct identities across the epithelial barrier and the immune system. The secondary dataset communicated by Yoshida et al. was filtered for acutely infected adults for whom nasal swab data were available and comprised 14 patients with mild to severe COVID-19 ($n_{cells} = 49,185$)[25].

In the T cell cluster from the primary dataset, the most abundantly expressed transcripts among patients with COVID-19 were those derived from *IFNG* ($n_{donors} = 20$, $f_{cells} = 31\%$), followed by *TNF* ($n_{donors} = 20$, $f_{cells} = 16\%$), *FASLG* ($n_{donors} = 18$, $f_{cells} = 13\%$), *CD40LG* ($n_{donors} = 12$, $f_{cells} = 3\%$), and, less frequently, *IL2*, *IL10*, and *IL21* (Supplementary Fig. 7). Transcripts encoding cytotoxic effector molecules were also detected, including *PRF1* ($n_{donors} = 22$, $f_{cells} = 27\%$), *GZMA* ($n_{donors} = 18$, $f_{cells} = 30\%$), and *GZMB* ($n_{donors} = 19$, $f_{cells} = 30\%$) (Supplementary Fig. 7). A comparable pattern was detected in the secondary dataset, although fewer cells expressed *IFNG* mRNA (11%). All relevant data obtained from T cells originally located in the infected upper airways epithelium are provided in Supplementary Dataset 7.

Collectively, these analyses showed that genes encoding cytotoxic and other effector molecules, including IFN-γ, were expressed frequently among T cells isolated from the upper airways of patients with mild to severe COVID-19.

### T cell expression of mRNA encoding IFN-γ in the upper airways is linked with antigen presentation and viral control during acute infection with SARS-CoV-2

In line with these findings, a previous study reported that acutely infected nasopharyngeal tissue harbored T cells expressing *IFNG* mRNA, likely reflecting specificity for SARS-CoV-2[26]. We therefore identified responders (Ziegler, $n = 18$; Yoshida, $n = 10$) and non-responders (Ziegler, $n = 16$; Yoshida, $n = 4$) among patients with mild to severe COVID-19, defined as those with or without *IFNG* mRNA+ T cells, respectively. Further interrogation of the primary dataset segregated by responder status revealed that 16 of the 32 initially annotated cell subsets contained DEGs ($P_{adj.} < 0.05$, absolute logfold change [LFC] > 0.25). The highest numbers of upregulated DEGs were present in developing and *FOXJ1*high ciliated cells ($n = 352$ for both) (Supplementary Dataset 1), which are abundant in the nasopharynx

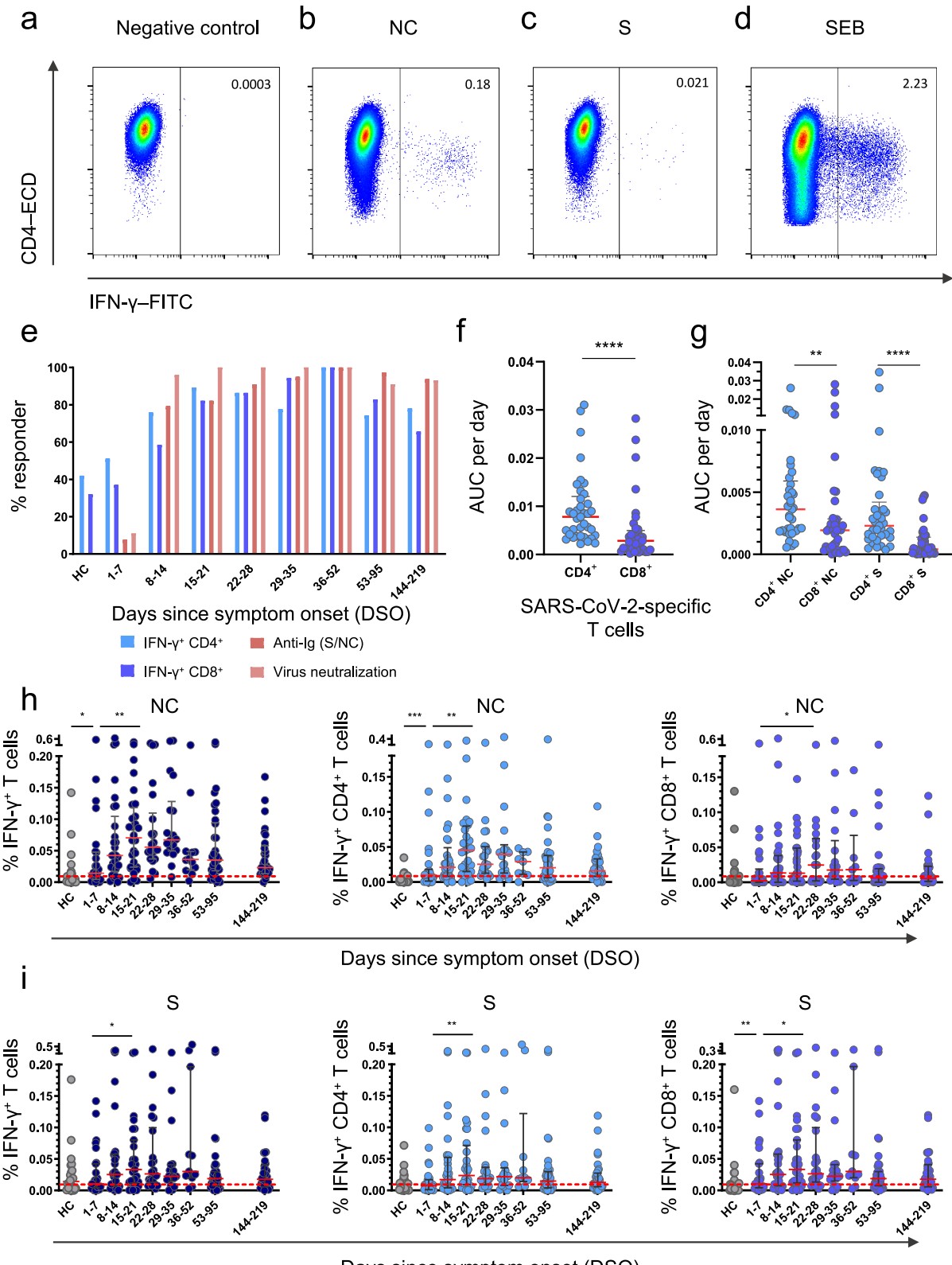

and frequent targets of SARS-CoV-2[24]. In responders, these cells overexpressed master transcription factors involved in antiviral immunity, such as *STAT1* and *IRF1*, and genes associated with antigen processing and presentation, such as *HLA-A*, *HLA-B*, *HLA-C*, *HLA-E*, and *TAP1* (Fig. 5a). Many of these genes are regulated by *IRF1*. Multiple HLA class 1 and class II genes were also upregulated among ciliated cells

from responders in the secondary dataset, alongside *STAT1* and *TAP1* (Fig. 5b and Supplementary Dataset 4). In addition, ciliated cells from responders overexpressed several proteasome subunits in both datasets, and other less abundant target cell types in the upper airways displayed similar patterns of gene expression. Consequently, genes associated with antigen processing and presentation were

**Fig. 2 | T cell responses against the nucleocapsid and spike proteins of SARS-CoV-2. a–d** Representative flow cytometry plots showing the identification of IFN-γ⁺ CD4⁺ T cells in the absence of stimulation (**a**) or in the presence of overlapping nucleocapsid (NC) peptides (**b**), overlapping spike (S) peptides (**c**), or staphylococcal enterotoxin B (SEB) as the positive control (**d**). Plots are gated on CD3. Numbers indicate the percent frequency of CD4⁺ T cells that produced IFN-γ. **e** Responder frequencies for IFN-γ⁺ CD4⁺ and IFN-γ⁺ CD8⁺ T cells specific for NC or S, antibody titers against NC or S, and antibody-mediated neutralization of SARS-CoV-2 (HC, healthy control). **f, g** Area under the curve (AUC) per day comparisons of the overall magnitude of SARS-CoV-2-specific CD4⁺ versus CD8⁺ T cells (**f**) and the overall magnitude of SARS-CoV-2-specific CD4⁺ versus CD8⁺ T cells broken down by target protein (NC versus S). Each dot represents one donor. **h** Frequencies of all NC-specific T cells (left), NC-specific CD4⁺ T cells (middle), and NC-specific CD8⁺

T cells (right). Each dot represents one donor. The cutoff is indicated by the dotted red line. **i** Frequencies of all S-specific T cells (left), S-specific CD4⁺ T cells (middle), and S-specific CD8⁺ T cells (right). Each dot represents one donor. The cutoff is indicated by the dotted red line. Data are shown as median ± IQR (**f, g, h, i**). Sample sizes in (**e, h, i**): HC, n = 24; 1–7 DSO, n = 25; 8–14 DSO, n = 30; 15–21 DSO, n = 28; 22–28 DSO, n = 20; 29–35 DSO, n = 18; 36–52 DSO, n = 8; 53–95 DSO, n = 34; 144–219 DSO, n = 29. Sample size in (**f, g**): n = 37. P values in (**f**): ****P < 0.0001; (**g**): ***P = 0.0005, ****P < 0.0001; (**h**): NC-specific IFN-γ⁺ T cells: *P = 0.017, **P = 0.0032; NC-specific IFN-γ⁺ CD4⁺ T cells: **P = 0.0018, ***P = 0.0005; NC-specific IFN-γ⁺ CD8⁺ T cells: *P = 0.042; (**i**): S-specific IFN-γ⁺ T cells: *P = 0.038; S-specific IFN-γ⁺ CD4⁺ T cells: **P = 0.0063; S-specific IFN-γ⁺ CD8⁺ T cells: *P = 0.038, **P = 0.0085 (Mann–Whitney U test or Wilcoxon signed rank test, two-sided). Source data are provided as a source data file.

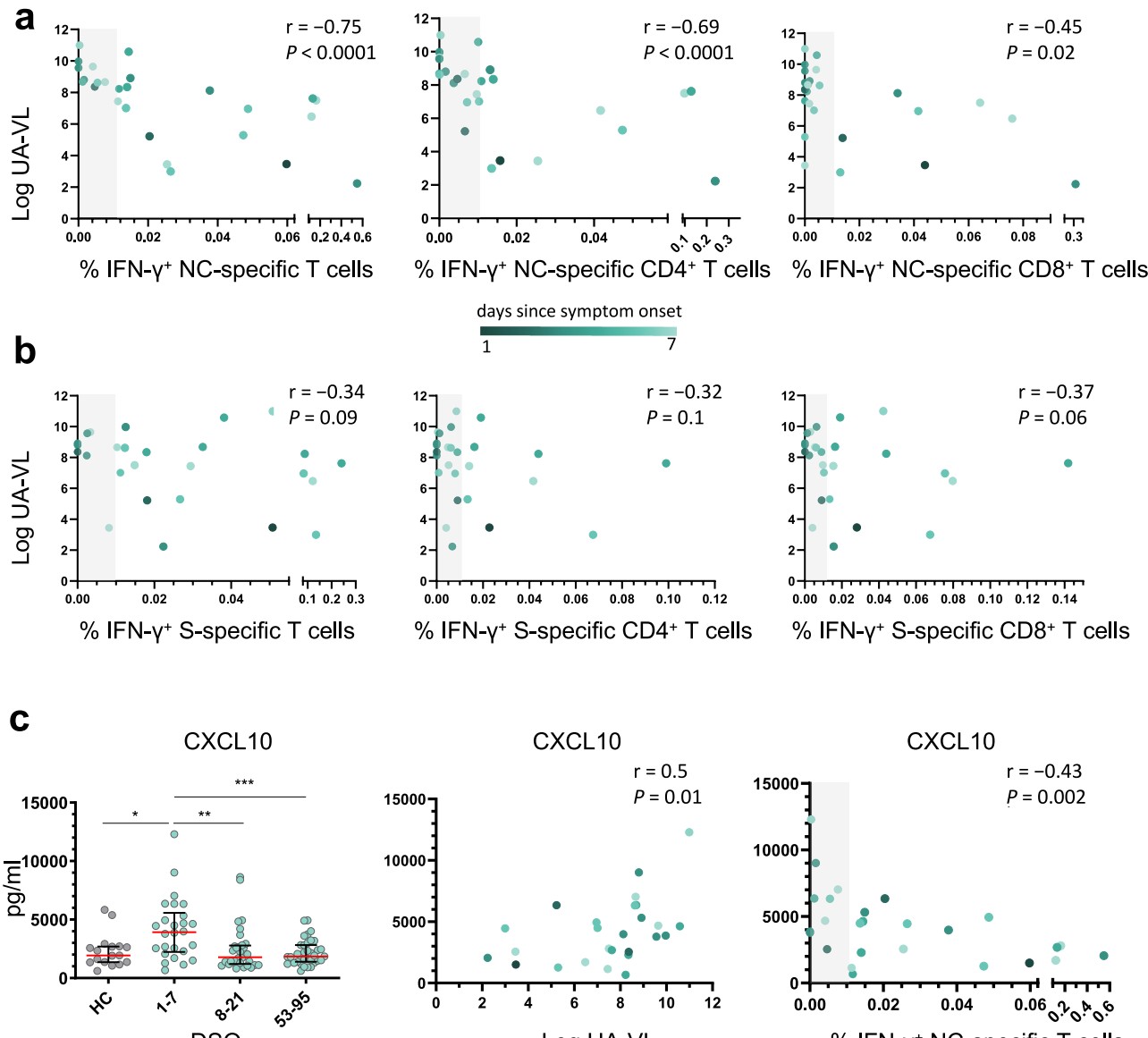

**Fig. 3 | Nucleocapsid-specific T cell responses correlate inversely with upper airways viral loads and systemic markers of inflammation during acute infection with SARS-CoV-2. a, b** Spearman rank correlations showing upper airways viral loads (UA-VLs) versus the frequencies of all NC-specific T cells (left), NC-specific CD4⁺ T cells (middle), or NC-specific CD8⁺ T cells (right) (**a**, n = 25) and the frequencies of all S-specific T cells (left), S-specific CD4⁺ T cells (middle), or S-specific CD8⁺ T cells (right) (**b**, n = 25) during the first week after symptom onset. **c** Left: plasma concentrations of CXCL10 are shown for healthy controls (HCs,

n = 17) and longitudinally for patients according to the number of days since symptom onset (1–7 DSO, n = 25; 8–21 DSO, n = 32; 53–95 DSO, n = 35). *P = 0.01, **P = 0.003, ***P = 0.0007 (Mann–Whitney U test, two-sided). The green scale stratifies patients according to days since symptom onset at presentation. Data are shown as median ± IQR. Middle and right: Spearman rank correlations showing plasma concentrations of CXCL10 during the first week after symptom onset versus UA-VLs (middle) and the frequencies of all NC-specific T cells (right). The gray bar indicates non-responders (right). Source data are provided as a source data file.

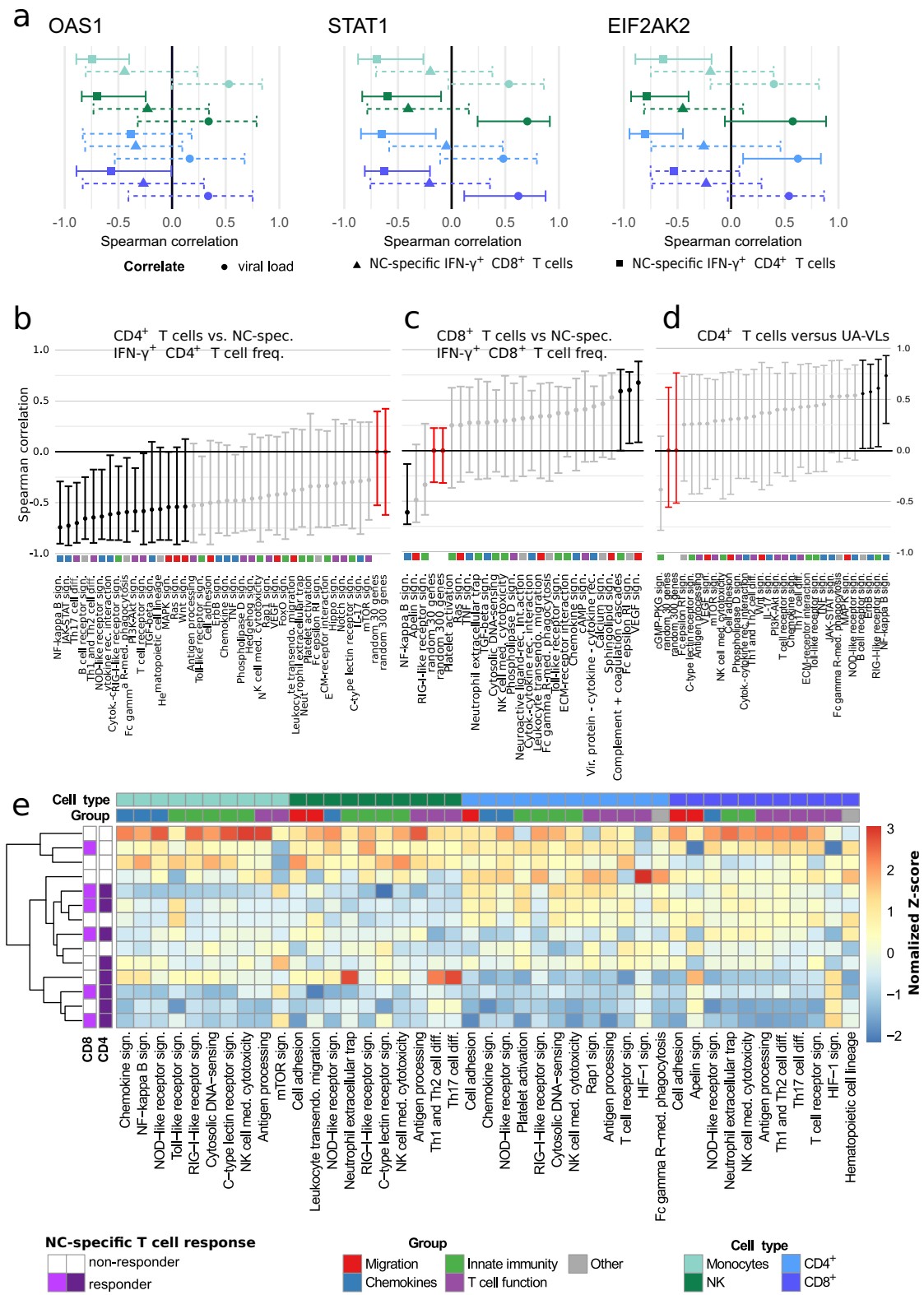

significantly enriched among ciliated cells from responders across multiple gene ontology (GO) terms, as were genes associated with signaling via type I and type II IFNs (Fig. 5c and Supplementary Datasets 3 and 6). Some differences between the datasets were also notable. For example, overexpressed marker genes in the primary dataset included *IRAK1*, *IRAK3*, and *FOS*, whereas overexpressed marker genes in the secondary dataset included *EIF3AK2*, *OAS1-3*,

*IFITM1*, *IFITM3*, *IFIT3*, *IFIT1*, and *IFI44*, which encode proteins with antiviral effector functions[12,27–30].

Similar enrichments were observed for ciliated cells in pathway analyses aligned to the KEGG database (Supplementary Datasets 2 and 5). Moreover, enriched pathways among ciliated cells from responders exhibited high combined scores for apoptosis, cellular senescence, necroptosis, and signaling via TNF. In the primary dataset,

**Fig. 4 | Gene expression profiles in immune cell subsets during acute infection with SARS-CoV-2.** RNA sequencing data were obtained from circulating CD4⁺ T cells (light blue), CD8⁺ T cells (dark blue), monocytes (light green), and NK cells (dark green) isolated during the first week after symptom onset (*n* = 14 patients with mild COVID-19). **a** Spearman rank correlations showing mean expression scores for *OAS1* (left), *STAT1* (middle), and *EIF2AK2* (right) versus NC-specific IFN-γ⁺ CD4⁺ (squares) and NC-specific IFN-γ⁺ CD8⁺ T cell frequencies (triangles) and upper airways viral loads (circles, UA-VLs). Whiskers show 95% confidence intervals calculated using bootstrapping with replacement using sample numbers equal to the original dataset. Solid lines indicate significance. Dashed lines indicate correlation results below the threshold for significance. **b–d** Spearman rank correlations showing mean pathway gene expression scores for CD4⁺ T cells versus NC-specific IFN-γ⁺ CD4⁺ T cell frequencies (**b**), CD8⁺ T cells versus NC-

specific IFN-γ⁺ CD8⁺ T cell frequencies (**c**), and monocytes versus UA-VLs. Whiskers as in (**a**). Data are shown as *r* values with 95% confidence intervals. Black and light grey lines indicate significant and non-significant associations, respectively. Red lines indicate reference control *r* values derived from 30 or 300 random genes as shown. Colored squares indicate the pathway group (manual annotation). **e** Spearman rank correlations for all KEGG pathways in the categories *Signal Transduction*, *Signaling Molecules and Interaction*, *Immune System*, and *Cell Growth and Death*. Data are shown as z-normalized mean pathway expression scores. Patients were clustered by expression profile similarity. Pathways are shown for cell subsets with significant enrichment scores in patients versus healthy controls (top row, *P* < 0.05; exact *P* values are provided in Supplementary Tables 1–4). Source data are provided as a source data file.

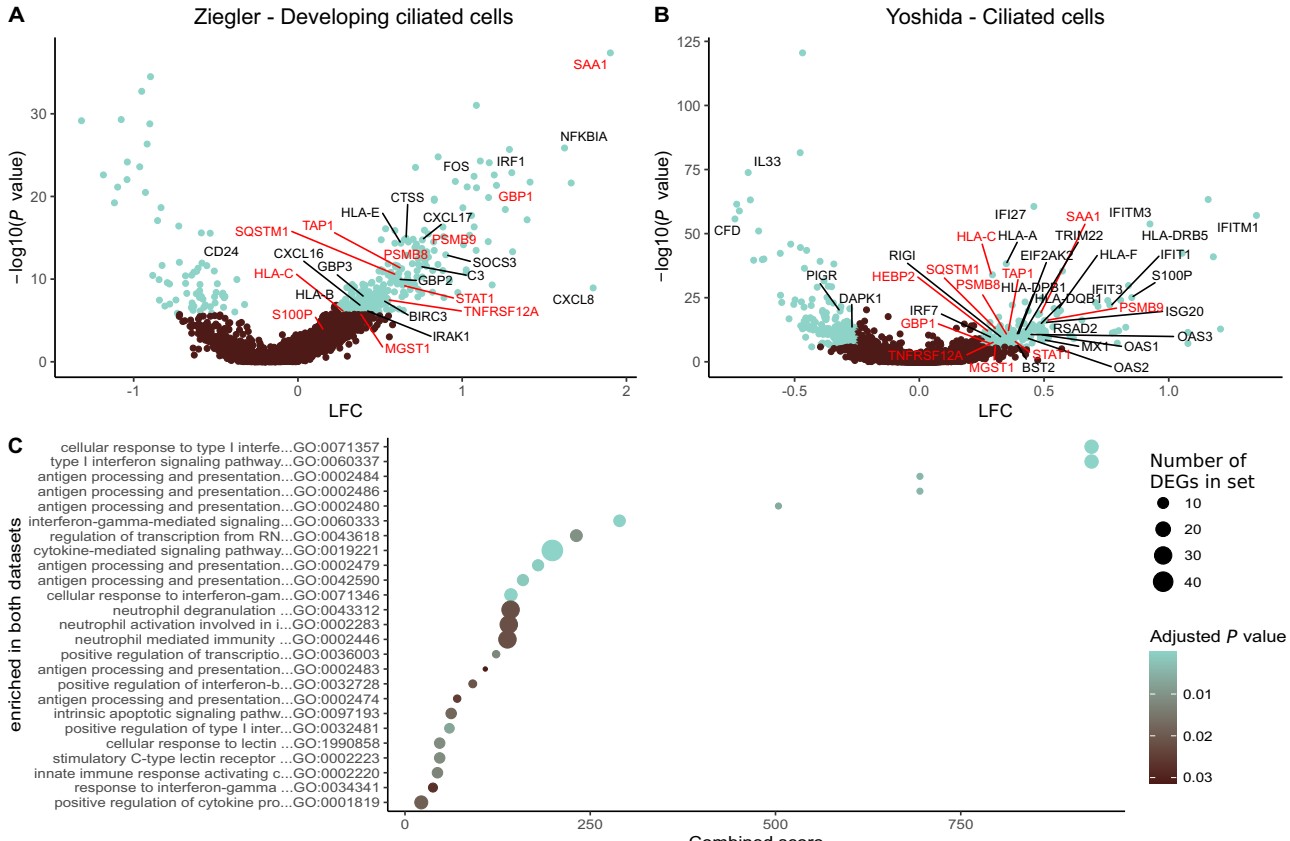

**Fig. 5 | The presence of T cells expressing mRNA encoding IFN-γ in the upper airways is linked with the upregulation of genes associated with antigen processing and presentation. a, b** Volcano plots showing DEGs (blue; *P*adj. < 0.05, absolute LFC > 0.25) among the biggest clusters of ciliated cells from responders in the primary (**a**) and secondary (**b**) datasets. Genes annotated in red are significant in both datasets, and genes annotated in black are significant in one dataset. **c** Gene

ontology (GO) terms enrichment plot for pathways significantly enriched in both datasets (*P*adj. < 0.05). Dot size represents the average number of significant DEGs that contributed to the term, and dot color represents the adjusted *P* value (*P*adj.). *X*-axis shows combined scores as reported by enrichR. Source data are provided as supplementary datasets.

we also found that responders exhibited higher fractions of SARS-CoV-2 RNA-free cells and lower abundances of SARS-CoV-2 RNA in infected cells compared with non-responders (responders, *n*cells = 11,871; non-responders, *n*cells = 5386; *P* = 0.00013), thereby aligning our results with biological efficacy (Supplementary Fig. 8). This analysis was not performed on the secondary dataset, because patients numbers were low and often lacked values for viral RNA.

Collectively, these findings indicated that the presence of T cells expressing mRNA encoding IFN-γ in the upper airways was associated with enhanced target cell conditioning for immune recognition, globally upregulated viral clearance mechanisms, and better localized control of SARS-CoV-2.

## Discussion

In this study, we undertook a comprehensive evaluation of adaptive immune responses, inflammatory cascades, and gene expression profiles among circulating immune cell subsets to define the correlates of viral control during acute infection with SARS-CoV-2. We found that genetic and plasma markers of systemic inflammation peaked during the first week after symptom onset and correlated directly with UA-VLs, whereas the contemporaneous frequencies of circulating viral NC-specific CD4⁺ and CD8⁺ T cells correlated inversely with various inflammatory markers and UA-VLs. Moreover, we identified high frequencies of activated CD4⁺ and CD8⁺ T cells in acutely infected naso-pharyngeal tissue, many of which expressed genes encoding various

effector molecules, such as cytotoxic proteins and IFN-γ. The presence of *IFNG* mRNA⁺ T cells in the infected epithelium was further linked with common patterns of gene expression among virus-susceptible target cells and better local control of SARS-CoV-2. Collectively, these results indicated a protective role for viral NC-specific T cells during the acute phase of infection with SARS-CoV-2, thereby providing an immune correlate that could inform the development of more effective vaccines against COVID-19.

T cells have been implicated as mediators of immune protection in some but not all studies of acute infection with SARS-CoV-2[7–11]. These discrepancies may relate to the exact timing of sample acquisition. In our study, the inverse correlation between circulating viral NC-specific T cell frequencies and UA-VLs was apparent only during the first week after symptom onset, prior to seroconversion. At this time, many of our patients exhibited high plasma concentrations of proinflammatory cytokines, many of which have been linked previously with severe disease, including the CXCR3 ligand CXCL10[8,31,32]. In line with an earlier study[8], we detected an inverse correlation between the frequencies of circulating viral NC-specific T cells and plasma concentrations of CXCL10, which in turn correlated directly with UA-VLs. Similar relationships were observed for NF-κB signaling pathway gene expression scores among circulating immune cell subsets, hinting at a potential mechanism. Indeed, many cytokines are transactivated via the NF-κB signaling pathway, including those implicated previously in the inflammatory storm that accompanies severe COVID-19, such as IL-1, IL-6, IL-8, TNF, and CXCL10[33]. These results supported the notion that immune control of early viral replication attenuates the local and systemic inflammation characteristic of severe COVID-19[34].

Unexposed individuals frequently harbor cross-reactive T cells with functional specificity for SARS-CoV-2, which likely arise in the memory pool as a consequence of previous infections with other viruses that exhibit a degree of structural homology, such as CCCVs[19–21]. In our study, all patients were seropositive for one or more CCCVs before the emergence of detectable antibody responses against SARS-CoV-2, and many healthy controls exhibited T cell cross-reactivity against S (54%) and NC (21%). However, it should be noted that amino acid sequence conservation between CCCVs and SARS-CoV-2 is rather limited across NC (< 30%) and that de novo priming of antiviral T cells from the naive pool could have occurred before clinical presentation[35].

In line with our finding that viral NC-specific but not viral S-specific T cell frequencies correlated inversely with UA-VLs, another study reported that cross-reactive viral NC-specific but not viral S-specific T cells appeared to protect exposed contacts from infection with SARS-CoV-2[36]. Previous work has also identified broad T cell reactivity against the major viral Gag proteins (matrix, capsid, and NC) but not the viral Env protein as a correlate of immune protection against HIV-1[37,38]. This observation could be explained by the rapid processing and presentation of Gag epitopes prior to viral integration and de novo gene expression[39]. In this context, it is notable that the corresponding virions are known to contain substantially higher amounts of NC compared with S or Env, respectively, and that target cells infected with SARS-CoV-2 in vitro have been shown to express approximately fivefold more NC compared with S[40–42]. NC has also been found abundantly in ex vivo analyses of upper airways target cells infected with SARS-CoV-2[24]. It is further notable here that viral matrix-specific and NC-specific T cell responses have been associated with protection against disease and reduced viral shedding after influenza virus infection[14]. The abundant expression of internal viral proteins may therefore facilitate early antigen presentation at surface densities sufficient to trigger cognate T cells more rapidly than external viral proteins, leading to greater immune efficacy. This paradigm makes sense in the context of our study and cautions against vaccine strategies that immunize solely against the S protein of SARS-CoV-2.

*IFNG* mRNA⁺ T cells were common in acutely infected nasopharyngeal tissue, likely as a consequence of viral antigen recognition via the TCR[26]. Moreover, the presence of nasopharyngeal *IFNG* mRNA⁺ T cells was associated with distinct patterns of gene expression among site-matched target cells, which upregulated pathways associated with antigen processing and presentation, apoptosis regulation, and innate antiviral responses, and also less frequently harbored SARS-CoV-2 RNA. In line with these findings, which suggested a coordinated network of viral suppression mechanisms driven by the influx of *IFNG* mRNA⁺ T cells during acute infection, nasopharyngeal target cells in responders also expressed lower amounts of SARS-CoV-2 RNA.

Several preclinical studies have provided support for the notion that next-generation vaccines would benefit from the inclusion of NC antigens to enhance immune efficacy against SARS-CoV-2. For example, IFN-γ production by viral NC-specific T cells in the airways was found to be a key determinant of outcome in mice infected with influenza virus or SARS-CoV-1[12,43], and local immunization with a single conserved NC epitope recognized by CD4⁺ T cells was sufficient to protect mice from MERS or SARS-CoV-1[12]. Intranasal vaccination of cynomolgus macaques with structural proteins from the inner virion core has also been shown to induce potent NC-specific T cell immunity and reduce peak UA-VLs by almost two orders of magnitude in the absence of neutralizing antibody responses against SARS-CoV-2[44]. Moreover, convalescent patients have been shown to harbor tissue-resident memory T cells targeting the most immunogenic regions of SARS-CoV-2, including epitopes derived from NC[16], consistent with a role in protection against recurrent episodes of COVID-19[45,46].

There are several limitations to our study. First, our cohort was relatively small and did not include patients with severe COVID-19. Second, we only report correlations, precluding a definitive assessment of antiviral efficacy. Third, we were unable to define antigen specificity in the single-cell RNA sequencing datasets, instead relying on the expression of *IFNG* mRNA as a surrogate marker of T cell activation driven by cognate engagement with epitopes derived from SARS-CoV-2[26]. Fourth, overlapping peptide sets can be suboptimal for the detection of functional CD8⁺ T cell responses, albeit with the concomitant advantage of global antigenic coverage[47,48]. Fifth, responders typically harbored higher overall frequencies of T cells in the infected epithelium compared with non-responders, potentially reflecting enhanced immune cell recruitment and/or other phenomena with possible impacts on viral replication. In spite of these caveats, our results provided clear evidence of a protective role for viral NC-specific T cells in the context of acute infection with SARS-CoV-2, thereby arguing for inclusion of the corresponding antigens in next-generation vaccines designed to combat COVID-19.

## Methods

### Ethics
Written informed consent was obtained from all participants in accordance with the principles of the Declaration of Helsinki. This study was approved by the Ethics Committee of the Faculty of Medicine at LMU Munich (20–371).

### Study participants
A total of 37 patients with acute COVID-19 were recruited into this study between May and December 2020 under the umbrella of the longitudinal KoCo19 Study[49]. All participants tested positive for SARS-CoV-2 via RT-PCR. At the time of recruitment, only the Wuhan strain (lineage A) was circulating in Germany. Clinical presentation was assessed using the WHO Clinical Progression Scale. All patients in this study had mild symptoms that did not require hospitalization and therefore scored a maximum of 3[18]. Healthy controls were recruited prior to vaccination and tested negative for SARS-CoV-2 via RT-PCR.

## Upper airways viral loads

Nasopharyngeal viral loads were quantified as described previously[49]. Briefly, RT-PCR was performed using a TANBead Maelstrom 9600 (Taiwan Advanced Nanotech Inc.) with an OptiPure Viral Auto Plate Kit (Taiwan Advanced Nanotech Inc.). SARS-CoV-2 RNA was quantified using an Allplex 209-nCov Assay (SeeGene) with a STARlet IVD (See-Gene). UA-VLs were calculated using standardized dilutions of SARS-CoV-2 RNA (INSTAND).

## Antibody titers

SARS-CoV-2-specific antibodies were assayed in EDTA plasma as described previously[50,51] using the following kits: Anti-SARS-CoV-2-ELISA Anti-S1 IgA (EI-S1-IgA, Euroimmun), Anti-SARS-CoV-2-ELISA Anti-S1 IgG (EI-S1-IgG, Euroimmun), and Elecsys Anti-SARS-CoV-2 Anti-N (Ro-N-Ig, Roche).

## Neutralization assays

Pseudotyped viral particles were generated via cotransfection of HEK 293 T cells (BEI Resources #NR-52511) with plasmids encoding HIV-1 Tat, HIV-1 Gag/Pol, HIV-1 Rev, luciferase, and the S protein of SARS-CoV-2 (Wu01 S, EPI_ISL_406716 lacking the cytoplasmic domain) using the FuGENE 6 Transfection Reagent (Promega). Culture supernatants were harvested at 48 h and 72 h after transfection, passed through a filter (pore size = 0.45 μm), and stored at −80 °C. Viral titers were established by infecting ACE2-expressing 293 T cells as described previously[52]. Luciferase activity was revealed after 48 h via the addition of luciferin/lysis buffer (10 mM MgCl$_2$, 0.3 mM ATP, 0.5 mM coenzyme A, 17 mM IGEPAL, and 1 mM D-luciferin in Tris-HCl) and measured using a Tristar Microplate Reader (Berthold Technologies). Neutralization assays were performed using serum samples as described previously[53]. Briefly, serial dilutions of serum were incubated with pseudovirus supernatants for 1 h at 37 °C. ACE2-expressing 293 T cells were then added in 15 μg/ml polybrene and incubated for a further 48 h at 37 °C. Luciferase activity was determined as above. Results were expressed for each sample as the 50% inhibitory dilution (ID$_{50}$) after subtraction of background relative light units (RLUs). ID$_{50}$ values were calculated using a non-linear fit model to plot agonist versus normalized dose-response curves with variable slopes in Prism version 7 (GraphPad). Samples that did not achieve 50% neutralization (serum ID$_{50}$ < 10) were assigned a value halfway below the lower limit of quantification (serum ID$_{50}$ = 5).

## Common cold coronavirus serology

Antibodies against the common cold coronaviruses 229E, NL63, OC43, and HKU1 were assayed in CPDA plasma using a recomLine SARS-CoV-2 IgG Kit (Mikrogen Diagnostik #7374).

## Flow cytometry

PBMCs were isolated within 6 h of blood collection via density gradient centrifugation (Cytiva Sweden AB), resuspended in RPMI 1640 medium (Thermo Fisher Scientific #61870-010) supplemented with 1% penicillin/streptomycin and 10% fetal calf serum (complete medium), and then stimulated immediately with 15mer peptide pools overlapping by 11 amino acids representing the NC or S proteins of SARS-CoV-2 (1 μg/ml/peptide, Miltenyi Biotec Peptivator SARS-CoV-2 Prot_N #130-126-699 or Prot_S #130-1226-701) for 16 h at 37 °C in the presence of anti-CD28 (clone L293, 1 μg/ml, BD Biosciences #340975), anti-CD49d (clone L25, 1 μg/ml, BD Biosciences #340976), and brefeldin A (5 μg/ml, Sigma-Aldrich). Negative control wells lacked stimulants (complete medium alone), and positive control wells contained staphylococcal enterotoxin B (SEB, 0.6 μg/ml, Sigma-Aldrich #11100-45-1). Cells were then stained with anti-CD4−ECD (clone SFCI12T4D11, Beckman Coulter #6604727), anti-CD8−APC-AF750 (clone B9.11, Beckman Coulter #A94683), anti-CD57−APC (clone QA17A04, BioLegend #303306), anti-PD1−PE-Cy5.5 (clone PD1.3, Beckman Coulter

#B36123), and anti-CXCR5−PE-Cy7 (clone J252D4, BioLegend #356924). Labeled cells were fixed/permeabilized using a FoxP3/Transcription Factor Staining Buffer Set (eBioscience #00-52123-43, #00-8333-56, and #00-5223-56) and further stained intracellularly with anti-CD3−APC-AF700 (clone UCHT1, Beckman Coulter #B10823), anti-IFN-γ−FITC (clone B27, BioLegend #506504), anti-IL2−PE (clone MQ1-17H12, BioLegend #500307), anti-TNF-α−BV510 (clone mAb11, BioLegend #502950), anti-CTLA-4−BV421 (clone BNI3, BioLegend #369606), anti-Ki-67−BV605 (clone Ki-67, BioLegend #350522), and anti-CD40L−BV785 (clone 24-31, BioLegend #310842). Samples were acquired using a CytoFLEX Flow Cytometer (Beckman Coulter). Data analysis was performed using FlowJo software version 10 (FlowJo LLC). SARS-CoV-2-specific T cell responses were defined on the basis of IFN-γ production and were considered positive at a frequency of ≥0.01% after background subtraction if greater than the corresponding unstimulated values by a factor of ≥2.

## Plasma cytokines and proteins

Concentrations of CCL2, CCL3, CCL4, CCL5, CCL17, CCL19, CD23, CXCL1, CXCL4, CXCL5, CXCL10, CXCL11, galectin-1, galectin-3, galectin-9, Gas6, ICAM-1, IL-2, IL-4, IL-10, IL-19, MICA, NCAM-1, PD-L1, syndecan-1, and TFPI were determined in CPDA plasma using a customized 26-plex marker panel (R&D Systems) as described previously[54]. Sample plates were read the same day using a Luminex MAGPIX (Thermo Fisher Scientific).

## RNA sequencing

Libraries were prepared from immune cell subsets (n = 500 cells each) using the Prime-seq protocol[55], and quality was determined using a High Sensitivity DNA Kit (Agilent Bioanalyzer). Paired-end sequencing (150 bp) was performed using an S1 or an S4 flow cell on a NovaSeq System (Illumina). An average of $1 \times 10^7$ reads were acquired per subset per sample. Preprocessing and quantification of the raw data was conducted using zUMIs[56] and referenced against GENCODE V35. Further analyses were performed using non-normalized outputs that mapped to exonic regions only (full data). Raw inputs were normalized using DESeq2 version 1.36.0[57]. Analyses were limited to participants in the KoCo19 study enrolled within the first week of symptom onset (n = 14) and healthy controls (n = 8). Initial pathway enrichment analyses were performed using R package gage version 2.46.0[58]. Pathways were included from the KEGG database mapped to BRITE terms in the groups *Signal Transduction* and *Signaling Molecules and Interaction* (environmental information processing), *Immune System* (organismal systems), and *Cell Growth and Death* (cellular processing). ENSEMBL IDs were used in the original dataset and converted to Entrez IDs using the org.Hs.eg.db R package version 3.15.0[59]. ID mappings for some genes were non-existent or not unique. The relevant genes were discarded in the former case or assigned to the first match in the latter case. Spearman's formula was used to calculate correlations among gene/pathway expression, cell type frequencies, and UA-VLs. Normalized read counts were used for individual genes, and average expression of composite genes was used for pathways. A confidence interval was calculated using boot-strapping of the original data by random resampling with replacement to estimate the range of possible correlations, with subsequent calculation of the mean expression score for each relevant pathway. Reference pathways were generated from 30 (smallest size) or 300 random genes (biggest size). Bootstrapping was performed over 1,000 iterations for each pathway. Correlation coefficients were then ordered and used to pick intervals at quantile values of 2.5% (low) and 97.5% (high).

## Statistics

Basic statistical analyses were performed using non-parametric tests in Prism version 8 (GraphPad).

## Analysis of single-cell RNA sequencing data

The primary dataset from Ziegler et al. was acquired from the Single Cell Portal (https://singlecell.broadinstitute.org/single_cell/study/SCP1289/). The secondary dataset from Yoshida et al. was acquired from the COVID-19 Cell Atlas (https://www.covid19cellatlas.org/). Data were normalized using Seurat version 4.1.0[60]. One patient was excluded from the primary dataset due to the presence of abnormally high numbers of macrophages (patient 19). Analyses were performed using the author-provided 'Detailed Cell Annotation'. T cells with at least one RNA read mapping to a selected function were classified as function-positive. Differentially expressed genes and pathways in the *IFNG*+ and *IFNG*− patient groups were identified using the FindMarkers function with default settings in Seurat version 4.1.0. Each previously reported cluster in the original annotation[24] was interrogated with no initial cutoff for LFC. All remaining clusters were used for reference. Pathway and GO term analyses were based on marker genes with an LFC of 0.25 in either direction and a *P* value of <0.05. Enrichment analyses were performed using enrichR (Kuleshov et al.[61]). Pathway analyses were limited to the following BRITE categories: *Signal Transduction*, *Signaling Molecules and Interaction*, *Immune System*, and *Cell Growth and Death*. Identical analyses were performed on the secondary dataset using the author-provided 'Annotation Level 2'. The common logarithm of SARS-CoV-2 total corrected RNA reported previously[24] was used to quantify host cell VLs. Patient groups were assigned as above. Values from all cells in the *IFNG*+ and *IFNG*− groups formed the test distribution for the *IFNG*+ and *IFNG*− groups, and comparisons were performed using a two-sided Mann-Whitney U test. Similar results were obtained using uncorrected read counts for SARS-CoV-2 RNA.

## Interaction models

A univariate linear mixed effects model was established using the default settings in CensReg[62]. Point estimates for the model parameters were obtained by minimizing the negative log-likelihood function using numerical minimization. Standard errors were derived from the inverse of the Hessian matrix evaluated at the point estimates. The likelihood function was constructed using truncated conditional normal distributions based on normality assumptions about individual effects and error terms to account for the limits of viral detection. A mixed effects model was also used to solidify the observed relationship as a correlation between a score for the subset of pathways and cell fractions and/or VLs. A second mixed model equation was added using Julia for joint modeling of subsets and VLs. This model included VL as a mediator of additional confounders to evaluate the influence of the true non-censored VL on each pathway score, despite the censored structure of the observed VLs. The outer marginalization of random effects within the likelihood was approximated using Gauss-Hermite quadrature[63], with weights obtained via the Julia package FastGaussQuadrature across 10 quadrature points (https://juliaapproximation.github.io/FastGaussQuadrature.jl/stable/). Gradients were obtained using automatic differentiation in the Julia package ForwardDiff[64]. Pathways were prefiltered by running ordinary least squares regressions to determine those potentially influenced by the VL. Data preprocessing was conducted in Python using Pandas[65] and NumPy[66]. All code is publicly available at https://github.com/manuhuth/early_t_cell_control.git.

## Reporting summary

Further information on research design is available in the Nature Portfolio Reporting Summary linked to this article.

## Data availability

The data are not openly available as they are subject to human data protection regulations. However, data will be made available upon reasonable request to the corresponding author (geldmacher@lrz.uni-muenchen.de). Single-cell RNA datasets used in this publication have been published previously by Ziegler et al. (https://singlecell.broadinstitute.org) and Yoshida et al. (https://covid19.cog.sanger.ac.uk). Data points in the figures are included in the published source data files or in the supplementary datasets. Source data are provided with this paper.

## Code availability

The codes used in Figs. 4, 5 and Supplementary Figs. 7, 8 are available online at https://github.com/TropI-LMU/Eser2022. All code for computational modeling is publicly available at https://github.com/manuhuth/early_t_cell_control.git.

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

## Acknowledgements

We wholeheartedly thank all study participants for their time, patience, trust, and constancy. We are grateful for financial support from the Bavarian State Ministry of Science and the Arts (KoCo19, M.Ho.; Long-term follow-up of cellular and humoral immune responses after SARS-CoV-2 infection and vaccination, C.G.), the Deutsche Forschungsgemeinschaft (GE 2128/3-1 and HO 2228/12-1; J.H., M.Ho., and C.G.), and the Ministry for Education and Research (01KI20271, M.Ho.). D.A.P. was supported by the National Institute for Health Research (COV-LT2-0041). Various kits and machines were kindly provided at discounted rates by Euroimmun, Roche, and Mikrogen. This study would not have been possible without the passionate contribution of staff in the Division of Infectious Diseases and Tropical Medicine at the University Hospital, LMU Munich, as well as medical students involved in the field work. Part of this work has been done for the doctoral theses of T.M.E., M.Hu., F.D., and K.Pe. Members of the KoCo19 study group: Mohamed Ibraheem Mohamed Ahmed, Emad Alamoudi, Jared Anderson, Maximilian Baumann, Marc Becker, Marieke Behlen, Jessica Beyerl, Rebecca Böhnlein, Isabel Brand, Anna Brauer, Vera Britz, Jan Bruger, Friedrich Caroli, Lorenzo Contento, Max Diefenbach, Jana Diekmannshemke, Paulina Diepers, Anna Do, Gerhard Dobler, Jürgen Durner, Ute Eberle, Judith Eckstein, Tabea Eser, Volker Fingerle, Felix Forster, Turid Frahnow, Jonathan Frese, Günter Fröschl, Christiane Fuchs, Mercè Garí, Otto Geisenberger, Leonard Gilberg, Kristina Gillig, Philipp Girl, Arlett Heiber, Christian Hinske, Janna Hoefflin, Tim Hofberger, Michael Höfinger, Larissa Hofmann, Sacha Horn, Kristina Huber, Christian Janke, Ursula Kappl, Charlotte Kiani, Arne Kroidl, Michael Laxy, Ronan Le Gleut, Reiner Leidl, Felix Lindner, Silke Martin, Rebecca Mayrhofer, Anna-Maria Mekota, Dafni Metaxa, Hannah Müller, Katharina Müller, Leonie Pattard, Michel Pletschette, Michael Pritsch, Stephan Prückner, Konstantin Pusl, Peter Pütz, Ernst-Markus Quenzel, Katja Radon, Elba Raimúndez, Camila Rothe, Nicole Schäfer, Yannik Schälte, Paul Schandelmaier, Lara Schneider, Sophie Schultz, Mirjam Schunk, Lars Schwettmann, Heidi Seibold, Peter Sothmann, Paul Stapor, Fabian Theis, Verena Thiel, Sophie Thiesbrummel, Niklas Thur, Julia Waibel, Claudia Wallrauch, Franz Weinauer, Simon Winter, Julia Wolff, Pia Wullinger, Houda Yaqine, Sabine Zange, Eleftheria Zeggini, Anna Zielka, Thomas Zimmermann.

## Author contributions

T.M.E. performed most of the experiments and analyzed data supported by M.I.M.A., F.D., K.H., K.Pe., R.R.A., N.C., and A.W.; O.B., A.L., L.N., and K.Pu. analyzed data; M.Hu. and J.H. performed computational modeling; K.H. and L.L. performed HLA typing; G.P. performed viral sequencing; M.B. and D.A.P. contributed editorially and intellectually; J.R., P.F., A.M., L.O., I.K., and M.Ho. contributed samples; K.V. and F.K. measured neutralizing antibodies; L.O., A.W., I.K., M.Ho., and C.G. conceived the study and wrote the clinical protocol; J.H., M.Ho., and C.G. acquired funding; T.M.E., O.B., and C.G. wrote the manuscript with input from all contributors.

## Funding

Open Access funding was enabled and organized by Projekt DEAL.

## Competing interests

The authors declare no competing interests.

## Additional information

[1]Division of Infectious Diseases and Tropical Medicine, University Hospital, LMU Munich, 81377 Munich, Germany. [2]German Center for Infection Research (DZIF), Partner Site Munich, 81377 Munich, Germany. [3]Institute of Computational Biology, Helmholtz Zentrum München, 85764 Neuherberg, Germany. [4]Center for Mathematics, Technische Universität München, 85748 Garching, Germany. [5]Department of Medicine I, University Hospital, LMU Munich, 81377 Munich, Germany. [6]German Center for Cardiovascular Research (DZHK), Partner Site Munich Heart Alliance, 81377 Munich, Germany. [7]Institute of Infection and Global Health, University of Liverpool, Liverpool L69 2BE, UK. [8]Center for Infectious Medicine, Department of Medicine Huddinge, Karolinska Institutet, Karolinska University Hospital Huddinge, 141 86 Stockholm, Sweden. [9]Division of Infection and Immunity, Cardiff University School of Medicine, University Hospital, Heath Park, Cardiff CF14 4XN, UK. [10]Systems Immunity Research Institute, Cardiff University School of Medicine, University Hospital, Heath Park, Cardiff CF14 4XN, UK. [11]Laboratory of Experimental Immunology, Institute of Virology, Faculty of Medicine and University Hospital Cologne, University of Cologne, 50931 Cologne, Germany. [12]German Center for Infection Research (DZIF), Partner Site Bonn-Cologne, 50937 Cologne, Germany. [13]Center for Molecular Medicine Cologne (CMMC), University of Cologne, 50931 Cologne, Germany. [14]Max von Pettenkofer Institute for Hygiene and Medical Microbiology, LMU Munich, 81377 Munich, Germany. [15]Faculty of Mathematics and Natural Sciences, University of Bonn, 53113 Bonn, Germany. ✉e-mail: geldmacher@lrz.uni-muenchen.de

