## [Peer Review File · Nature Communications]

Reviewers' comments:

Reviewer #1 (Remarks to the Author):

In this study Eser et al. analysed nasopharyngeal and peripheral blood samples for SARS-CoV-2 specific cellular and humoral immune responses. They performed serology on blood plasma samples to detect SARS-CoV-2 and common cold coronavirus antibodies and performed a CBA to measure cytokines. Nasopharyngeal swabs were used to measure viral loads. PBMCs were stimulated with peptides derived from the nucleocapsid or spike proteins and T cell activation was measured by flow cytometry for intracellular IFN- γ . Frequencies of activated T cells were correlated with viral loads. Inverse correlations were observed for NC-specific total, CD4+ and CD8+ T cells, whereas there were no significant correlations for S-specific T cells. Bulk RNA sequencing was performed on CD4+ T cells, CD8+ T cells, NK cells and monocytes derived from patients with acute but mild COVID-19. Gene expression profiles were linked to donors' in vitro CD4+ and CD8+ T cell response after peptide stimulation. The authors found that there was a negative correlation between donors' nucleocapsid-specific CD4+ T cell response and OAS1, STAT1 and EIF2AK2 gene expression which replicated previously reported findings. They also find that there are IFNG expressing T cells in nasopharyngeal samples from acutely infected individuals.

Specific comments:

1. Can the authors analyse TCR usage from the RNAseq data to potentially identify the specificity? If there is a TCR bias, it could hint at a potential epitope that the T cells are responding to, based on what is known in the literature on SARS-CoV-2-specific TCRs.

2. When discussing the RNA sequencing data, can the authors more clearly represent that it is the IFNG gene, and not the IFN- γ protein that they observed in the T cells. It can be slightly confusing since the authors also refer to IFN- γ + T cells from the in vitro peptide stimulations.

3. The authors commonly refer to NC-specific CD4+ and CD8+ T cells, however this is based on in vitro stimulation of PBMCs. Have the authors confirmed the specificity of the cytokine-producing T cells with pMHC tetramers?

4. The conclusion the authors make from Figure 5 data is not quite accurate, as they state "cytotoxic and effector molecules were expressed..." in lines 298-300, however the analysis is on RNA transcripts, therefore it would require that protein expression was also assessed to make this conclusion. Do the authors have protein level data to show the ex vivo phenotype?

5. The authors state that the NC- and S-specific CD4+ and CD8+ T cell response peaked at 3 or 4 weeks post symptom onset, respectively. However, the statistical analysis does not support that there is a significant increase in the IFN- γ response at these timepoints. Can the authors clarify how they came to this conclusion?

6. It looks like the clustering in Fig 4E is more so defined by the donors' CD8+ T cell responder status rather than the CD4+ T cells as the text currently states. I.e. The top cluster only has non-responders, middle cluster is mixed, and bottom cluster is only responders. Whereas each cluster has a mix of CD4+ T cell responders. Could the authors reassess their interpretation of the unsupervised clustering and address it in terms of the CD8+ response. Could the authors also better label each cluster as the results text currently is ambiguous with statements like "the other cluster..." "another cluster..." Maybe label as cluster 1, 2, 3.

7. The authors state in lines 193-196 that NC-specific T cells are mediators of viral clearance in the upper airways, but their T cell data is derived from PBMCs. Can the authors comment on how their data support this conclusion?

8. Could the authors comment on how the gate in Supp Fig 6 was drawn and if it has any significance. Is this determined mathematically or drawn by eye? Is it necessary if the cells of interest are already colored as red or blue.

9. Could the authors please include the sample size (n=) in the figure legends for each group shown in each graph.

10. A legend is needed for the different shades of green in Fig 1A.

11. Could the authors determine if the x-axis labels "35+" and "60+" for DSO in Fig 1B and Supp Fig 1, Fig 2H and I, etc. could be changed to "36-59" and "60-179" as the value of 35 is already stated in the "29-35" group and stating e.g. 35+ is not technically correct if excluding the 60+ and 180+ from the 35+ group.

12. Duplicate FACS plots are shown in Fig 2B and 2C. Could the authors please show the correct FACS plot for each condition.

13. Could the authors please provide a full gating strategy for their flow cytometry analysis

14. Could the authors please provide the raw antibody titres used to generate Supp Fig 2 and address how the high, intermediate and low groups were determined?

15. Could the authors make the green color scale for DSO in Fig 3 more obvious and have 1 at the bottom and 7 at the top, it is slightly misleading that the higher number is at the bottom of the scale.

16. A legend is needed for the solid and dashed lines in Figure 4A.

17. Could the authors please make nomenclature for EIF2AK2/PKR consistent between the main text and the Fig 4A right graph.

18. A legend is needed for the black, grey and red lines in Fig 4B-D.

19. Supp Fig 5 legend is cut off on bottom of the page.

20. Could the authors include viral load data in the hierarchical clustering analysis as an additional parameter to determine if there is an immune cell gene signature that associates with higher/lower loads. Additionally, the WHO severity score could be another factor to include in this.

21. Could the authors provide more information on what peptide pools were used from Miltenyi. What was the length of the peptides and were they overlapping, and what medium was used for the peptide stimulations?

22. Could the authors add more labelling to Figure 4 B-D graphs to directly state which correlation each graph is referring to in the figure as panel B and C compare activated vs total CD4+/CD8+ T cells, but panel D is comparing monocytes with viral loads. It would be helpful to have this clearly labelled in the figure. Additionally, it would be informative to show the raw values used to do this correlation analysis.

23. Figure 4E needs a label on the heatmap colour scale

Reviewer #2 (Remarks to the Author):

In the research report “Early nucleocapsid-specific T cell responses associated with control of SARS-CoV-2 in the upper airways and reduced systemic inflammation before seroconversion” by Eser and colleagues the authors identify the association of nucleocapsid (NC) specific T cells and viral control in early SARS-CoV2 infection. The authors first measure upper airway viral load (UA-VL) using qPCR and measure the development of SARS-CoV2 neutralizing antibodies in serum over time. They then measure T cell response to NC and S derived peptide in PBMC. In a correlative analysis they demonstrate that the fraction of NC reactive but not S protein reactive T cells is associated with decreased UA-VL as a surrogate for disease severity. Using multiplexed cytokine measurements as a surrogate for disease severity, they further show that CXCL10 and other inflammatory response cytokines are inversely correlated with the fraction of NC reactive T cells and correlate with UA-VL. This prompts the authors to investigate the transcriptional response of circulating immune subsets in this patient cohort using bulk RNA-seq. In a targeted analysis they find that OAS1, STAT1 and EIF2AK2 expression in CD4, CD8, NK and monocytes are inversely correlated with the frequency of NC reactive CD4, while the same markers are positively correlated with UA-VL. They then perform KEGG pathway analysis and identify pathways with correlating/anticorrelating with NC-reactive CD4, CD8 and UA-LV. Inflammatory pathways correlated with UA-VLs but anticorrelated with T cell frequency. T cell effector pathways were positively correlated with NC reactive CD8 T cells. To conclude the RNA-seq analysis the authors performed hierarchical clustering and identified 3 clusters across the for RNA-seq data sets This analysis shows that responders (defined NC-reactive T cell positive) show lower expression of many inflammatory pathways across all immune populations. The authors conclude that this suggests that NC specific CD4/CD8 T cells mitigate the systemic inflammatory response through enhanced clearance of the infection. To investigate the tissue response at the site of infection the authors then turn to published scRNA data of cells obtained through nasopharyngeal swab of healthy or SARS-CoV2 infected individuals with different disease severity. They reannotate the T cell compartment and stratify patients based on IFNg gene expression as responders (likely to have NC reactive T cells) and non-responders akin to the stratification of their original cohort. Based on this stratification they find that a significant number of cell clusters contain differentially expressed genes between the two groups. They focus on differentially expressed genes and pathways enriched in cells annotated as developing ciliated cells, interferon-responsive ciliated cells and CD8 T cells among others and find increased IFNg regulated pathways in responders (viral response, antigen presentation etc.).

In summary, this paper reports a role of NC reactive T cells for disease control. This is based on immunological studies of an original cohort of patients with mild/moderate COVID-19. This is also reflected in the general transcriptional landscape in PBMC subpopulations from patients of this original cohort. The extension of these findings to tissue immunity based on the reanalysis of published single-cell data relies on strong assumptions that are not supported by the data. These either need additional experiments (beyond the scope of a revision) or more applicable datasets. However, in light of recent published data at nature communications I don't feel that this is necessary to be of general interest to the field as the data presented is complimentary to a recent studies on NC specific T cell response, e.g,

Kundu, 2022, Nat. Comms., <https://doi.org/10.1038/s41467-021-27674-x> demonstrating presence of NC (cross)reactive but not S reactive T cells in household members of COVID+ patients prevent them from becoming sick. If major concerns below can be addressed by the authors, the study would be of interest for the general readership.

Major comments:

- Statistics for serological response (Figure S2) not provided. It looks like the difference for NL63 is distinct among +/- patients and responders/non responders. I would suggest normalizing the data to 1 for each comparison and then comparing the fraction of sero-negative patients in responders and non-responders followed by contingency analysis and fisher's exact test to support the statement in the main text. This is important as it might suggest some pre-existing immunity to other coronaviruses contributing to a more robust NC specific response in patients with mild disease.

- The analysis of the single cell data has various limitations. In its current form I am unable to added value beyond the analysis performed in the original publication. It is not necessarily required if conclusions on tissue immunity are toned down appropriately.

o First, the data set under investigation only provides very few T cells, the focus of this study. There are multiple data sets of airway samples available that are enriched for CD45+ cells (e.g. Szabo, 2021, Immunity, <https://pubmed.ncbi.nlm.nih.gov/33765436>; Bronchoalveolar lavage).

o Second, cell clustering is not demonstrated and the assignment of Cells to CD4/CD8 compartment is not shown. "Gating" on CD4 t cells in a scatter plot is rather unorthodox and separation could be demonstrated by e.g. looking at bimodal distribution of CD4 and CD8A/B transcripts. More importantly patient origin for the cells needs to be demonstrated. On average each patient only contributes ~10 CD4 cells and ~14 CD8 T cells. Since cytokines such as IFNg, IL2 etc. suffer from substantial dropout in single-cell data hence their absence needs to be interpreted with caution. Are patients with a "response" showing IFNg expression homogeneously responding?

o The number of "responders" vs "non-responders" (7+9) suggests that this is only based on patients with CD4 T cells (n = 16). With the number of cells (n=66) the analysis is not powered to make generalizable conclusions.

Minor comments:

- Data for S reactive T cells and cytokine response is not shown. This doesn't need to be presented as a figure but both S and NC response should also be shown as a supplementary table. Is this data corrected for multiple hypothesis testing?

- Axis labels and color legends in some figures of the single cell analysis missing making the figures less accessible and hard to interpret. Examples include the color of charts in Figure 6A and E, Axis in S6. In general additional legend would make the figures more accessible (e.g. figure 4A)

Reviewers' comments:

Reviewer #1 (Remarks to the Author):

In this study Eser et al. analysed nasopharyngeal and peripheral blood samples for SARS-CoV-2 specific cellular and humoral immune responses. They performed serology on blood plasma samples to detect SARS-CoV-2 and common cold coronavirus antibodies and performed a CBA to measure cytokines. Nasopharyngeal swabs were used to measure viral loads. PBMCs were stimulated with peptides derived from the nucleocapsid or spike proteins and T cell activation was measured by flow cytometry for intracellular IFN- γ . Frequencies of activated T cells were correlated with viral loads. Inverse correlations were observed for NC-specific total, CD4+ and CD8+ T cells, whereas there were no significant correlations for S-specific T cells. Bulk RNA sequencing was performed on CD4+ T cells, CD8+ T cells, NK cells and monocytes derived from

patients with acute but mild COVID-19. Gene expression profiles were linked to donors' in vitro CD4+ and CD8+ T cell response after peptide stimulation. The authors found that there was a negative correlation between donors' nucleocapsid-specific CD4+ T cell response and OAS1, STAT1 and EIF2AK2 gene expression which replicated previously reported findings. They also find that there are IFNG expressing T cells in nasopharyngeal samples from acutely infected individuals.

Response: We extremely thankful to the reviewer for this comprehensive review and many useful suggestions and criticism. Our core finding was that Nucleocapsid-specific T cell responses inversely correlated with upper airway viral loads and systemic inflammation during the first week of symptom onset before seroconversion. This review also pointed out, where the manuscript may not have been sufficiently clear. This is important, as we feel that some of the comments do not adequately reflect what was written or stated in the manuscript. One example is our finding that Nucleocapsid-specific CD4 and CD8 T cell responses negatively correlated with OAS1, STAT1 and EIF2AK2 gene expression. This finding does not at all replicate a previously reported finding, as stated by the reviewer (without providing a reference). We decided to specifically study mRNA expression of these genes in peripheral blood in relation to Nucleocapsid-specific T cell responses in peripheral blood in our cohort, because Zhao and colleagues showed in a murine vaccine model for SARS-CoV-1 that airway expression of IFN γ by T cells resulted in the upregulation of several IFN-related genes within airways of SARS-CoV-1 challenged, Nucleocapsid vaccinated mice, including STAT-1, PKR, and OAS-1, which are important for viral clearance (Zhao et al. 2016 *Immunity: Airway memory CD4+ T cells mediate protective immunity against emerging respiratory coronaviruses*).

Our presented data and analyses are therefore unique and novel and do not at all replicate previous findings. We did however recognized that particularly the labeling and legend for figure 4 was not clear on several key aspects of our analyses – correlations with NC-specific T cell frequencies. We therefore modified the relevant labeling to make more clear that the correlation analyses relates to NC-specific CD4 and CD8 T cell frequencies detected during the first week in peripheral blood and the gene expression profiles within sorted immune cell subsets. In the revised version we address all these points to improve the manuscript's clarity including figure 4.

Specific comments:

Reviewer 1: Can the authors analyse TCR usage from the RNAseq data to potentially identify the specificity? If there is a TCR bias, it could hint at a potential epitope that the T cells are responding to, based on what is known in the literature on SARS-CoV-2-specific TCRs.

Response: We thank the reviewer for his comment and indeed had similar thoughts. Unfortunately, this is not possible because standard untargeted scRNAseq data did not allow for analyses of TCR-CDR3 sequences.

It was however previously reported by Fischer et al. 2021 in Nature Communications (<https://doi.org/10.1038/s41467-021-24730-4>) that IFN- γ mRNA expression is specific for SARS-CoV-2-specific T cells upon cognate in vitro TCR stimulation. Further Fischer and colleagues found that tracheal aspirate CD8 T cells were overall very similar to SARS-CoV-2

reactive in peripheral blood and write the following on page 8, first paragraph: “While these tissue resident markers differentiated them from antigen-reactive peripheral blood CD8 T cells, tracheal aspirate CD8 T cells were overall very similar to reactive peripheral blood T cells—especially from the stimulated condition—with high expression of IFNG, PDCD1, or CD38 (Supplementary Fig. 10).”

We therefore feel that these results published by Fischer et al. 2022 support the validity of our approach of using IFN- γ mRNA expression in T cells from SARS-CoV-2 infected upper airway epithelia as a surrogate marker of functional activation in infected epithelium. In order to more prominently refer to this study, we added this reference on page 10 in the first sentence of the second paragraph:

„Acutely infected nasopharyngeal tissue harbours IFN- γ mRNA expressing T cells, likely reflecting specificity for SARS-CoV-2 (25). Next, we therefore identified responders (Ziegler: n = 18, Yoshida: n = 10) and non-responders (Ziegler: n = 16, Yoshida: n = 4) among the patients with mild to severe COVID-19, defined as those with or without IFN- γ transcript expressing T cells, respectively.”

and in the third paragraph page 13 of the discussion, where this reference was missing.

„IFN- γ mRNA⁺ T cells were common in acutely infected nasopharyngeal tissue, likely as a consequence of TCR stimulation by cognate SARS-CoV-2 antigens (25).“

Reviewer 1: When discussing the RNA sequencing data, can the authors more clearly represent that it is the IFNG gene, and not the IFN- γ protein that they observed in the T cells. It can be slightly confusing since the authors also refer to IFN- γ + T cells from the in vitro peptide stimulations.

Response: We have now adapted the relevant paragraph titles related to scRNA sequencing analyses of upper airway T cells to more clearly specify that our results relate to mRNA expression of IFN- γ and other cytokines or effector molecules.

Reviewer 1: The authors commonly refer to NC-specific CD4⁺ and CD8⁺ T cells, however this is based on in vitro stimulation of PBMCs. Have the authors confirmed the specificity of the cytokine-producing T cells with pMHC tetramers?

Response: We feel that there is no necessity to confirm specificity of our intracellular cytokine staining, as the detection antigen-specific CD4 and CD8 T cells upon vitro antigen stimulation of PBMCs and intracellular staining of IFN- γ is a very standard and specific method to detect antigen-specific T cells. Figure 2A-D show that IFN- γ is only expressed in SARS-CoV-2 antigen restimulated PBMCs, but not in the negative control (Figure 2A). This was the pattern seen for all stimulation experiments, with basically no relevant background staining in unstimulated negative controls.

In order to make this clearer also in the results (2nd section on page 5) we adapted the text as follows:

„T cell responses against the viral NC and S proteins were measured longitudinally using flow cytometry to detect the intracellular production of IFN- γ , **which identified SARS-CoV-2-specific CD4 and CD8 T cells with high specificity.** “

Reviewer: The conclusion the authors make from Figure 5 data is not quite accurate, as they state “cytotoxic and effector molecules were expressed...” in lines 298-300, however the analysis is on RNA transcripts, therefore it would require that protein expression was also assessed to make this conclusion. Do the authors have protein level data to show the ex vivo phenotype?

Response: This statement refers to results/conclusions from our single cell gene expression or differentially expressed genes. We have now modified the wording to avoid confusion between gene expression and protein expression as follows:

„Collectively, these analyses showed that mRNA transcripts for IFN- γ , cytotoxic and other effector molecules were expressed frequently among T cells isolated from the upper airway of patients with mild to severe COVID-19.“

Reviewer: The authors state that the NC- and S-specific CD4+ and CD8+ T cell response peaked at 3 or 4 weeks post symptom onset, respectively. However, the statistical analysis does not support that there is a significant increase in the IFN- γ response at these timepoints. Can the authors clarify how they came to this conclusion?

Response: We did not provide or mention any statistical analyses in the manuscript about whether or not the peak of the NC- and S-specific CD4+ and CD8+ T cell response is significantly higher compared to the first week of symptom onset (or any other time point or group), making it difficult for us to understand this comment.

The increases in antigen-specific T cells between week 1 and week 3 (or 4) are statistically significant. However, we did not include this information, as we felt it was not relevant for the key conclusions of the manuscript, which mostly emphasizes on findings related to immunological and virological events and data from the first week after symptom onset, before this peak occurred. In order to clarify that NC- and S-specific T cells peaked between 1521 days post symptom onset with significant differences to week 1 post symptom onset, we now added the statistical analyses and adapted figures 2H and 2I for NC- and S-specific CD4 and CD8 T cells accordingly and added this information on page 6 on the first paragraph:

„Peak NC- and S-specific T cell frequencies were significantly higher during week 3 compared to the first week of symptom onset (NC-specific T cells: 7-fold increased, $p = 0.003$; S-specific T cells: 3-fold increased $p = 0.04$)“.

We further added the p-value to the other sentences where needed in the revised manuscript.

Reviewer: It looks like the clustering in Fig 4E is more so defined by the donors' CD8+ T cell responder status rather than the CD4+ T cells as the text currently states. I.e. The top cluster only has non-responders, middle cluster is mixed, and bottom cluster is only responders.

Whereas each cluster has a mix of CD4+ T cell responders. Could the authors reassess their interpretation of the unsupervised clustering and address it in terms of the CD8+ response. Could the authors also better label each cluster as the results text currently is ambiguous with statements like “the other cluster...” “another cluster...” Maybe label as cluster 1, 2, 3.

Response: We thank the reviewer for point out this discrepancy between Figure 4E and the text. The labelling of T cell subsets in Figure 4E had accidentally been switched. It is “more defined” by the donors’ NC-specific CD4+ T cell responder status rather than the NC-specific CD8+ T cell responder status. We have corrected this error in the new figure 4 and further revised figure legend, labelling to improve clarity of this figure.

Reviewer: The authors state in lines 193-196 that NC-specific T cells are mediators of viral clearance in the upper airways, but their T cell data is derived from PBMCs. Can the authors comment on how their data support this conclusion?

Response: We would like to emphasize that our wording in the submitted manuscript is more careful as follows: "Collectively, these findings supported a role for early IFN- γ expressing NC-specific CD4⁺ and CD8⁺ T cells as mediators of viral clearance in the upper airways...".

This more careful statement is fully supported by the presented data, because we found that frequencies of circulating IFN- γ expressing NC-specific CD4⁺ and CD8⁺ T cells early after infection were inversely correlated with upper airway viral load and systemic inflammation.

In the next results sections, we then aimed to elucidate how T cells may mediate or contribute to viral control in upper airway epithelial cells. Using public scRNA data sets, We therefore studied expression of IFN- γ and other cytokine/effector molecule mRNAs in T cells in SARS-CoV-2 infected upper airway epithelia and – in the revised version – identified differentially expressed genes in acutely infected patients with or without IFN- γ mRNA expressing cells in two independent data sets. As mentioned above in the response to comment 1, Fischer et al. found that IFN γ mRNA expression is a “surrogate biomarker” for SARS-CoV-2-specific T cells in acutely infected nasopharyngeal tissue, supporting our strategy.

Reviewer: Could the authors comment on how the gate in Supp Fig 6 was drawn and if it has any significance. Is this determined mathematically or drawn by eye? Is it necessary if the cells of interest are already colored as red or blue.

Response: The intent of the solid border (“gate”) in supplementary figure 6 was apparently insufficiently described in the figure legend and questioned by both reviewers 1 and 2. There was no manual gating as stated by both reviewers, who misinterpreted the visual aid from the supplementary figure 6 as ‘gating’, i.e. the method of partitioning. As was described in the method section of first submitted version of the manuscript in lines 775-779, the partitioning of T cells in subsets was done in the state-of-the-art manner using clustering from Seurat package coupled with verification based on the silhouette plots. The annotation was based on T cell markers performed by another established tool (scCATCH).

We acknowledge that bioinformatic annotation of small specific T cell subsets by scRNA sequencing data is less robust as for example compared to manual gating of CD4 versus CD8

T cells using flow cytometric analyses. Because identification of specific T cell subsets is not essential for the main manuscript's conclusions, including those relating to differentially expressed genes in virus target cells in subjects with and without IFN- γ mRNA positive T cells in the upper airways, we decided to omit differentiation of CD4 and CD8 T cells through annotation in the revised manuscript and revised supplementary figure 6 accordingly. Further we now present analyses from two independent data sets.

Reviewer: Could the authors please include the sample size (n=) in the figure legends for each group shown in each graph.

Response: We have now added the sample size in the figure legends for each group shown in figures 1 to 4.

Reviewer: A legend is needed for the different shades of green in Fig 1A.

Response: We have now added following text to the figure legend to better describe what the different shades of green mean.

“ .The different shades of green indicate for each subject on which day after symptom onset the upper airway virus load was determined during the first study week.....”

Reviewer: Could the authors determine if the x-axis labels “35+” and “60+” for DSO in Fig 1B and Supp Fig 1, Fig 2H and I, etc. could be changed to “36-59” and “60-179” as the value of 35 is already stated in the “29-35” group and stating e.g. 35+ is not technically correct if excluding the 60+ and 180+ from the 35+ group.

Response: We have changed the labelling according to the reviewers recommendation.

Reviewer: Duplicate FACS plots are shown in Fig 2B and 2C. Could the authors please show the correct FACS plot for each condition.

Response: We apologize for this embarrassing mistake. We now show the correct plots from a single subject for in Figure 2 for each condition.

Reviewer: Could the authors please provide a full gating strategy for their flow cytometry analysis

Response: We provide the gating strategy for identification of SARS-CoV-2-specific T cells in the new supplementary figure 2.

Reviewer: Could the authors please provide the raw antibody titres used to generate Supp Fig 2 and address how the high, intermediate and low groups were determined?

Response: As described in the materials and methods, the recomLine SARS-CoV-2 IgG Kit (Mikrogen Diagnostik), was used to determine serostatus for common cold coronaviruses 229E, NL63, OC43. This assay does not result in titers, but rather signal strength. We have now added following text to the materials and methods:

„Antibodies against the common cold coronaviruses 229E, NL63, OC43, and HKU1 were assayed in CPDA plasma using a recomLine SARS-CoV-2 IgG Kit (Mikrogen Diagnostik). High, intermediate and low responses were determined according to manufacturer instructions by visual comparison to a reference control for each analyzed sample on the same test stripe.“

Of note, the former supplementary figure 2 is now supplementary figure 3

Reviewer: Could the authors make the green color scale for DSO in Fig 3 more obvious and have 1 at the bottom and 7 at the top, it is slightly misleading that the higher number is at the bottom of the scale.

Response: We have now replaced the abbreviation “DSO” by “days since symptom onset” and added the explanation to the figure legend similar to figure 1:

“... The different shades of green indicate for each subject on which day after symptom onset the upper airway virus load was determined during the first study week.“

Reviewer: A legend is needed for the solid and dashed lines in Figure 4A.

Response: We thank the reviewer for pointing out insufficient clarity of figure 4 legend and have now improved clarity of figure legend and labels. The solid lines indicate significant correlation, while dashed lines are used for the correlation results below the threshold. We apologize for this oversight and modified the figure legend accordingly.

Reviewer: Could the authors please make nomenclature for EIF2AK2/PKR consistent between the main text and the Fig 4A right graph.

Response: We have corrected the inconsistency in the new version of the manuscript.

Reviewer: A legend is needed for the black, grey and red lines in Fig 4B-D.

Response: Again, we thank the reviewer for pointing this out. The solid lines indicate significant correlations, while grey lines are below the threshold. Red line represents a correlation expected from a random gene set (i.e. showing the correlation expected by chance). This information is included into the figure legend in the new version of the manuscript.

Reviewer: Supp Fig 5 legend is cut off on bottom of the page.

Response: We have missed this issue during the submission process and should have the full legend in the resubmitted version.

Reviewer: Could the authors include viral load data in the hierarchical clustering analysis as an additional parameter to determine if there is an immune cell gene signature that

associates with higher/lower loads. Additionally, the WHO severity score could be another factor to include in this.

Response: This information was indeed included in previous versions of the figure, but we eventually removed it to avoid overloading the visualization. As figure 4 and the hierarchical clustering analysis is already complex, we would prefer not to show these data in Figure 4E.

Reviewer: Could the authors provide more information on what peptide pools were used from Miltenyi. What was the length of the peptides and were they overlapping, and what medium was used for the peptide stimulations?

Response: We have now included more detailed information on peptides and cell culture media in the materials and methods.

Reviewer: Could the authors add more labelling to Figure 4 B-D graphs to directly state which correlation each graph is referring to in the figure as panel B and C compare activated vs total CD4+/CD8+ T cells, but panel D is comparing monocytes with viral loads. It would be helpful to have this clearly labelled in the figure. Additionally, it would be informative to show the raw values used to do this correlation analysis.

Response: We now added the recommended labelling to figure 4 to improve clarity and make it more intuitive to understand this complex figure.

Reviewer: Figure 4E needs a label on the heatmap colour scale

Response: The heatmap label – “normalized Z-score” is now shown and further described in the figure legend.

Reviewer #2 (Remarks to the Author):

In the research report “Early nucleocapsid-specific T cell responses associated with control of SARS-CoV-2 in the upper airways and reduced systemic inflammation before seroconversion” by Eser and colleagues the authors identify the association of nucleocapsid (NC) specific T cells and viral control in early SARS-COV2 infection. The authors first measure upper airway viral load (UA-VL) using qPCR and measure the development of SARS-CoV2 neutralizing antibodies in serum over time. They then measure T cell response to NC and S derived peptide in PBMC. In a correlative analysis they demonstrate that the fraction of NC reactive but not S protein reactive T cells is associated with decreased UA-VL as a surrogate for disease severity. Using multiplexed cytokine measurements as a surrogate for disease severity, they further show that CXCL10 and other inflammatory response cytokines are inversely correlated with the fraction of NC reactive T cells and correlate with UA-VL. This prompts the authors to investigate the transcriptional response of circulating immune subsets in this patient cohort using bulk RNA-seq. In a targeted analysis they find that OAS1, STAT1 and EIF2AK2 expression in CD4, CD8, NK and monocytes are inversely correlated with the frequency of NC reactive CD4, while the same markers are positively correlated with UA-VL. They then perform KEGG pathway analysis and identify pathways with correlating/anticorrelating with NC-reactive CD4,

CD8 and UA-LV. Inflammatory pathways correlated with UA-VLs but anticorrelated with T cell frequency. T cell effector pathways were positively correlated with NC reactive CD8 T cells. To conclude the RNA-seq analysis the authors performed hierarchical clustering and identified 3 clusters across the for RNA-seq data sets. This analysis shows that responders (defined NC-reactive T cell positive) show lower expression of many inflammatory pathways across all immune populations. The authors conclude that this suggests that NC specific CD4/CD8 T cells mitigate the systemic inflammatory response through enhanced clearance of the infection. To investigate the tissue response at the site of infection the authors then turn to published scRNA data of cells obtained through nasopharyngeal swab of healthy or SARS-CoV2 infected individuals with different disease severity. They reannotate the T cell compartment and stratify patients based on IFN γ gene expression as responders (likely to have NC reactive T cells) and non-responders akin to the stratification of their original cohort. Based on this stratification they find that a significant number of cell clusters contain differentially expressed genes between the two groups. They focus on differentially expressed genes and pathways enriched in cells annotated as developing ciliated cells, interferon-responsive ciliated cells and CD8 T cells among others and find increased IFN γ regulated pathways in responders (viral response, antigen presentation etc.).

In summary, this paper reports a role of NC reactive T cells for disease control. This is based on immunological studies of an original cohort of patients with mild/moderate COVID-19. This is also reflected in the general transcriptional landscape in PBMC subpopulations from patients of this original cohort. The extension of these findings to tissue immunity based on the reanalysis of published single-cell data relies on strong assumptions that are not supported by the data. These either need additional experiments (beyond the scope of a revision) or more applicable datasets. However, in light of recent published data at nature communications I don't feel that this is necessary to be of general interest to the field as the data presented is complimentary to a recent studies on NC specific T cell response, e.g, Kundu, 2022, Nat. Comms., <https://doi.org/10.1038/s41467-021-27674-x> demonstrating presence of NC (cross)reactive but not S reactive T cells in household members of COVID+ patients prevent them from becoming sick. If major concerns below can be addressed by the authors, the study would be of interest for the general readership.

Response: This is a comprehensive and good summary of our core findings and approach to dissect the tissue T cell response to acute SARS-CoV-2 infection.

We also thank the reviewer for pointing us towards this interesting article by Kundu et al. that reports results complementary to our own. We now include these findings in the discussion. Nonetheless, we feel that there are important differences between these findings: Our core finding is that during early SARS-CoV-2 infection the fraction of circulating nucleocapsid reactive T cells inversely correlates with decreased UA-VL, and hence this represents a T cell correlate of upper airway virus control. This finding is complementary but fundamentally different from what was reported by Kundu et al. 2022, which reports higher frequencies of pre-existing cross-reactive nucleocapsid-specific IL-2-secreting memory T cells in contacts who remained PCR-negative despite exposure, when compared with those who convert to PCR-positive.

Reviewer #2

Major comments:

Reviewer #2: Statistics for serological response (Figure S2) not provided. It looks like the difference for NL63 is distinct among +/- patients and responders/non responders. I would suggest normalizing the data to 1 for each comparison and then comparing the fraction of sero-negative patients in responders and non-responders followed by contingency analysis and fisher's exact test to support the statement in the main text. This is important as it might suggest some pre-existing immunity to other coronaviruses contributing to a more robust NC specific response in patients with mild disease.

Response: Our data did not show a significant difference for NL63 seropositivity between subjects with and without SARS-CoV-2 NC-specific T cell responses in the first week since symptom onset. We have now added more detail to the respective supplementary figure legend.

Reviewer

#2:

- The analysis of the single cell data has various limitations. In its current form I am unable to added value beyond the analysis performed in the original publication. It is not necessarily required if conclusions on tissue immunity are toned down appropriately.

Response: We fully agree with the reviewer that the scRNA analyses are a non-essential add on to our core findings, but strongly feel that it is important to analyze the gene expression of key cytokine and anti-viral effector molecules of T cells within SARS-CoV-2 infected tissues. We then performed a differential gene expression analyses to better understand the differences in an infected microenvironment and within SARS-CoV-2 target cells that are associated with IFN- γ mRNA expressing T cells in these infected tissues. The results are novel and do not replicate at all any findings by Ziegler et al., which does not focus on airway T cells or specifically the expression of IFN- γ mRNA by T cells. As is described by Fisher et al. 2022 in Nature Communications, nasopharyngeal T cells expressing IFN- γ mRNA are enriched for SARS-CoV-2 specific T cells. Fischer and colleagues also show that IFN- γ mRNA is a marker of SARS-CoV-2-specific T cells upon cognate antigen in vitro TCR stimulation (<https://doi.org/10.1038/s41467-021-24730-4>). Further Fischer and colleagues found that tracheal aspirate CD8 T cells were overall very similar to SARS-CoV-2 reactive in peripheral blood and write the following (page 8, first paragraph of the article): "While these tissue resident markers differentiated them from antigen-reactive peripheral blood CD8 T cells, tracheal aspirate CD8 T cells were overall very similar to reactive peripheral blood T cells—especially from the stimulated condition—with high expression of IFNG, PDCD1, or CD38 (Supplementary Fig. 10)."

We agree that the part of the analysis that is concentrating solely on the CD4+ and CD8+ cells is less robust due to the low cell numbers. In the light of this valid comment we have decided to remove the re-annotation and analysis of "CD8+ T cells only" from the manuscript. Instead we use the entire T cell subset for the classification of the subjects into responders and non-responders. Expectedly the above modification of the analysis changes the exact fold-changes

and p-values, but the direction in which the results are pointing remains the same. We have adapted the conclusions of the manuscripts accordingly.

In summary, our manuscript was not clear on several key aspects of our scRNA analyses in infected nasopharyngeal tissue and both reviewers misunderstood figure 5 and associated result paragraphs in the text in one way or the other (please see below).

We therefore decided to completely revise the corresponding figure 5 and text paragraphs and extend and verify our analyses – as recommended by reviewer 2 – by analyses of a second public scRNA data set by Yoshida et al. 2022, which we found more suitable for that purpose, as compared to the data set recommended by reviewer 2.

Reviewer 2: First, the data set under investigation only provides very few T cells, the focus of this study. There are multiple data sets of airway samples available that are enriched for CD45+ cells (e.g. Szabo, 2021, Immunity, <https://pubmed.ncbi.nlm.nih.gov/33765436>; Bronchoalveolar lavage).

Response: We thank the editor for emphasising the importance of a different dataset and pointing out an example. We however refrained from analysing this dataset for following reasons: the patients in the dataset are fairly severely affected by COVID-19 and include patients that die from it. Additionally the death from COVID-19 is heavily confounded with age, hence making it a difficult relationship to disentangle. After mining the European Nucleotide Archive for COVID-19 specific scRNA datasets, we have decided to use the data by Yoshida et. al. as validation for the finding made with Ziegler et. al. data.

Reviewer 2: Second, cell clustering is not demonstrated and the assignment of Cells to CD4/CD8 compartment is not shown. “Gating” on CD4 t cells in a scatter plot is rather unorthodox and separation could be demonstrated by e.g. looking at bimodal distribution of CD4 and CD8A/B transcripts. More importantly patient origin for the cells needs to be demonstrated. On average each patient only contributes "10 CD4 cells and "14 CD8 T cells. Since cytokines such as IFN γ , IL2 etc. suffer from substantial dropout in single-cell data hence their absence needs to be interpreted with caution. Are patients with a “response” showing IFN γ expression homogeneously responding?

Response: We are sorry for the poor choice of visual aid and wording in supplementary figure 6 as it was misinterpreted by both reviewers. There was no manual gating and the solid line from sup. figure 6 was merely the visual aid encapsulating the clusters created by different means. As described in the method section of the initial manuscript in lines 775-779, the partitioning of T cells in clusters was done in the state-of-the-art manner using clustering from Seurat package coupled with optimization based on the silhouette plots. The annotation was based on T cell markers performed by another established tool (scCATCH).

As we acknowledge that clustering and annotation of such small subsets can be unreliable we have decided to refrain from the analysis of CD8+ T cells in subjects with IFN γ mRNA+ CD4 T cells in the new version of the manuscript.

Instead we have included an additional dataset and based the differential gene expression analyses on the entire T cell subset including more than a thousand T cells in both data sets. We are aware that IFN γ is a gene with low RNA counts and susceptible to dropouts and indeed subjects with detectable IFN γ mRNA positive T cells often also had higher frequencies of T cells in the infected epithelium, possibly reflecting increased immune cell recruitment or other phenomena. Nonetheless we detected a highly IFN-specific and IFN γ -specific transcriptome signature in the analysed SARS-CoV-2 target cells, indicating a major role of IFN γ mRNA positive T cells rather than total T cell frequencies. We have this point now in the discussion within the limitations section on page 14 as follows:

“Fourth, patients with detectable IFN- γ mRNA+ T cells also often had higher frequencies of T cells in the infected epithelium, possibly reflecting increased immune cell recruitment or other phenomena.”

Reviewer 2: The number of “responders” vs “non-responders” (7+9) suggests that this is only based on patients with CD4 T cells (n = 16). With the number of cells (n=66) the analysis is not powered to make generalizable conclusions.

Response: As briefly mentioned above, this only applies to the analysis of CD8+ T cells shown in the previous version of Figure 5B & C, right panel and have omitted this part from the manuscript as discussed above.

Minor comments:

Reviewer 2: Data for S reactive T cells and cytokine response is not shown. This doesn't need to be presented as a figure but both S and NC response should also be shown as a supplementary table. Is this data corrected for multiple hypothesis testing?

Response: We are not completely clear to which manuscript section or figure this comment relates to. Figure 2 shows the longitudinal data for S-reactive T cells producing IFN γ upon in vitro restimulation. We do not see the need for multiple testing for example to assess the significance of an increase in S- or N-reactive T cells after the first week since symptom onset, as the analyses of T cell dynamics did not require a big number of concurrent tests.

Reviewer 2: Axis labels and color legends in some figures of the single cell analysis missing making the figures less accessible and hard to interpret. Examples include the color of charts in Figure 6A and E, Axis in S6. In general additional legend would make the figures more accessible (e.g. figure 4A)

Response: We have improved the labeling in the new manuscript and are grateful for the reviewers attention to details.

REVIEWERS' COMMENTS

Reviewer #1 (Remarks to the Author):

The authors addressed my concerns

Reviewer #2 (Remarks to the Author):

I thank the authors for a comprehensive revision of the manuscript. All points were addressed or discussed to my satisfaction. The manuscript is of great interest to the community (in particular in the context of T cell targeted vaccines) and the general readership of Nature communications and I recommend publication of the revised manuscript.